# A Bayesian Bootstrap Framework for Mutual Information Neural Estimation: Bridging Classical Mutual Information Learning and Bayesian Nonparametric Learning

**Forough Fazeliasl**                                                *fazelias@ualberta.ca*
*Department of Mathematical and Statistical Sciences*
*University of Alberta, AB, Canada*

**Michael Minyi Zhang**                                          *mzhang18@hku.hk*
*Department of Statistics and Actuarial Science*
*University of Hong Kong Pok Fu Lam, Hong Kong*

**Linglong Kong**                                                    *lkong@ualberta.ca*
*Department of Mathematical and Statistical Sciences*
*University of Alberta, AB, Canada*

**Bei Jiang**                                                          *bei1@ualberta.ca*
*Department of Mathematical and Statistical Sciences*
*University of Alberta, AB, Canada*

**Reviewed on OpenReview:** *https://openreview.net/forum?id=mqGzGKXnFi*

## Abstract

In this work, we introduce a Bayesian bootstrap resampling framework for estimating mutual information (MI) via "mutual information neural estimation" (MINE), making MINE directly applicable in a Bayesian nonparametric learning (BNPL) framework. The resulting estimator shows low variability across batch sizes and high-dimensional settings, as demonstrated through extensive numerical studies. In particular, our proposed bootstrap version yields tighter and lower-variance estimates than the original MINE formulation, both theoretically and empirically. We further demonstrate its practical value in a downstream task by improving VAE-GAN training within BNPL, leading to higher-quality outputs. Beyond enabling MI-based BNPL, the proposed bootstrap estimator also performs competitively against leading frequentist state-of-the-art benchmarks. Overall, our findings establish the first principled framework for Bayesian bootstrap-based MI estimation and highlight its effectiveness as a reliable tool for future BNPL studies.

## 1 Introduction

Estimating and optimizing mutual information (MI) is important in many machine-learning problems, from representation learning to generative modeling. The MI is defined through a Kullback-Leibler (KL) divergence between the joint and marginal distributions to measure the dependency between two variables, making its direct optimization intractable in practice. For this reason, most practical methods use a variational lower bound (VLB) together with an auxiliary function, usually a discriminator that compares joint and marginal samples (Belghazi et al., 2018; Hjelm et al., 2018). Many studies have improved these VLBs to make estimations tighter and more stable, and using these bounds is now a standard approach in MI-based learning (Nguyen et al., 2010; Oord et al., 2018; Dorent et al., 2025).

However, these developments have been designed mainly for standard learning settings. Bayesian nonparametric learning (BNPL) is now growing as a flexible way to model complex data (Fong et al., 2019; Lyddon et al., 2018; 2019; Dellaporta et al., 2022; Fazeli-Asl et al., 2024), with the potential to provide posterior inference under model misspecification; yet none of the existing VLB-MI estimators can be used directly in MI-based BNPL procedures. BNPL relies on posterior sampling induced by randomized objective functions together with posterior bootstrap algorithms, whereas current MI estimators do not account for the underlying distribution being treated as a random probability measure. This gap leaves MI-based BNPL without a practical or compatible lower bound estimator.

In this paper, we fill this gap by introducing a Bayesian bootstrap framework to estimate the VLB of MI. The core idea is to apply Bayesian bootstrap resampling for Mutual Information Neural Estimation (MINE) (Belghazi et al., 2018), replacing uniform sample weights with Dirichlet-resampled weights. Beyond providing this structure that allows VLB optimization to be used in MI-based BNPL, which is the main contribution of this work, we also show that this Bayesian bootstrap produces even a tighter VLB compared with the original MINE. We further compare our estimator numerically with several state-of-the-art VLBs, including NWJ (Nguyen et al., 2010), CPC/InfoNCE (Oord et al., 2018), and JSD-LB (Dorent et al., 2025), and show that it is a strong competitor that offers improved stability and lower variance. To illustrate the practical value of our approach, we incorporate the proposed estimator into a learning problem already formulated within the BNPL framework, namely a VAE-GAN architecture from Fazeli-Asl & Zhang (2023). This provides a natural context to assess how our estimator performs in MI-based optimization and to demonstrate its effectiveness on real data. These results may encourage researchers to consider their learning problems within a Bayesian context.

The rest of the paper is organized as follows. Section 2 reviews standard VLB methods for MI estimation. Section 3 presents our main contribution, the Bayesian bootstrap framework DPMINE, and outlines its basic theoretical properties. Section 4 reports experiments showing that DPMINE provides stable, low-variance MI estimates and can also improve training in a BNP VAE-GAN model when applied within BNPL for data-synthesis purposes. Section 5 concludes with a brief discussion of future work. Additional proofs and experimental details appear in the Appendix.

## 2 Related Work

In this section, we briefly review standard VLBs for estimating MI. We then give a short note on their common use in standard data-generating tasks.

**Mutual Information Estimation.** Estimating MI in machine learning optimization problems is challenging, especially with high dimensional data or small mini-batches. Let $(\mathbf{U}, \mathbf{V})$ be a pair of random variables with a joint cumulative distribution function (CDF) $F_{\mathbf{UV}}$ and marginal CDFs $F_{\mathbf{U}}$ and $F_{\mathbf{V}}$, defined over the product space $\mathcal{U} \otimes \mathcal{V}$. The MI between $\mathbf{U}$ and $\mathbf{V}$ is given by:

$$\mathrm{MI}(\mathbf{U}, \mathbf{V}) = \mathrm{D}_{\mathrm{KL}}(F_{\mathbf{UV}}, F_{\mathbf{U}} \otimes F_{\mathbf{V}}) = H(\mathbf{U}) - H(\mathbf{U}|\mathbf{V}), \tag{1}$$

where $\mathrm{D}_{\mathrm{KL}}(\cdot, \cdot)$ denotes the KL divergence, and $H(\cdot)$ and $H(\cdot|\cdot)$ represent marginal and conditional Shannon entropies, respectively.

Standard approaches to MI estimation either depend on parametric assumptions on the data distribution that rarely hold or rely on nonparametric methods such as $k$-nearest neighbors (Al-Labadi et al., 2022). Such nonparametric approaches become unstable as the dimension increases due to the curse of dimensionality (Sugiyama, 2012). These limitations have led to neural VLBs to estimate MI, which scale better but can still struggle when the true MI is large or when training relies on small mini-batch sizes (Dorent et al., 2025).

**Standard Neural VLBs.** The KL-MINE introduced by Belghazi et al. (2018) is one of the most widely used neural VLBs for MI estimation. For instance, Belghazi et al. (2018) introduced KL-MINE, which uses the Donsker-Varadhan (DV) representation (Donsker & Varadhan, 1983) of the KL divergence to form a lower bound of equation 1 as:

$$L_{\boldsymbol{\gamma}}^{\mathrm{KL}}(\mathbf{U}, \mathbf{V}) = \mathbb{E}_{F_{\mathbf{UV}}}[T_{\boldsymbol{\gamma}}(\mathbf{U}, \mathbf{V})] - \ln \mathbb{E}_{F_{\mathbf{U}} \otimes F_{\mathbf{V}}}[e^{T_{\boldsymbol{\gamma}}(\mathbf{U}, \mathbf{V})}], \tag{2}$$

where $\{T_{\boldsymbol{\gamma}}\}_{\boldsymbol{\gamma} \in \boldsymbol{\Gamma}}$ be a set of continuous functions parameterized by a neural network on a compact domain $\boldsymbol{\Gamma}$ that maps $\mathcal{U} \otimes \mathcal{V}$ to $\mathbb{R}$ and $\ln(\cdot)$ denotes the natural logarithm. In this framework, the auxiliary statistic $T_{\boldsymbol{\gamma}}$ serves as the key component that learns to discriminate between samples from $F_{\mathbf{U}\mathbf{V}}$ and those from $F_{\mathbf{U}} \otimes F_{\mathbf{V}}$.

The KL-MINE is then obtained by maximizing VLB given in equation 2 over $\boldsymbol{\gamma} \in \boldsymbol{\Gamma}$. In practice, equation 2 is estimated using the empirical cumulative distribution functions (ECDFs), $F_{\mathbf{U}_{1:n}} := \frac{1}{n}\sum_{i=1}^{n}\delta_{\mathbf{U}_i}$ and $F_{\mathbf{V}_{1:n}} := \frac{1}{n}\sum_{i=1}^{n}\delta_{\mathbf{V}_i}$:

$$L_{\boldsymbol{\gamma}}^{\mathrm{KL}}(\mathbf{U}_{1:n}, \mathbf{V}_{1:n}) = \sum_{\ell=1}^{n}\frac{1}{n}T_{\boldsymbol{\gamma}}(\mathbf{U}_\ell, \mathbf{V}_\ell) - \ln\sum_{\ell=1}^{n}\frac{1}{n}e^{T_{\boldsymbol{\gamma}}(\mathbf{U}_\ell, \mathbf{V}_{\pi(\ell)})}, \tag{3}$$

where $\{\pi(1:n)\}$ denotes a random permutation of $\{1:n\}$, a standard technique to empirically approximate $\mathbb{E}_{F_{\mathbf{X}} \otimes F_{\mathbf{Y}}}(\cdot)$ in equation 2 (Belghazi et al., 2018; Hjelm et al., 2018). However, MINE is prone to high variances (Liao et al., 2020).

Another variant of MINE is based on a variational lower bound for the Jensen-Shannon (JS) divergence (Nowozin et al., 2016) between $F_{\mathbf{U}\mathbf{V}}$ and $F_{\mathbf{U}} \otimes F_{\mathbf{V}}$ (Hjelm et al., 2018; Jones et al., 2023). It is defined as

$$L_{\boldsymbol{\gamma}}^{\mathrm{JS}}(\mathbf{U}, \mathbf{V}) = \mathbb{E}_{F_{\mathbf{U}\mathbf{V}}}[-\zeta(-T_{\boldsymbol{\gamma}}(\mathbf{U}, \mathbf{V}))] - \mathbb{E}_{F_{\mathbf{U}} \otimes F_{\mathbf{V}}}[\zeta(T_{\boldsymbol{\gamma}}(\mathbf{U}, \mathbf{V}))], \tag{4}$$

where $\zeta(\cdot) = \ln(1 + \exp(\cdot))$ denotes the softplus function. Although JS-MINE yields lower variance and more stable results, its connection to the KL-based definition of MI remains unclear (Dorent et al., 2025).

Dorent et al. (2025) addresses this concern by providing a new VLB on the KL divergence expressed as a function of the JS divergence, clarifying why maximizing the JS divergence is equivalent to maximizing a lower bound on MI. Their VLB also overcomes well-known drawbacks of NWJ (Nguyen et al., 2010), which often has high variance, and CPC/InfoNCE (Oord et al., 2018), whose estimates are restricted by the batch size.

**Alternative Neural Bounds.** There are additional neural variational bounds for MI estimation, including VNMC and VPCE (Ivanova et al., 2024), which provide upper and lower bounds, respectively. These methods rely on additional modeling assumptions, including the availability of at least one distribution in closed form, and estimate MI using nested Monte Carlo procedures based on Bayes' rule. Consequently, they are commonly used in modeling settings that differ from distribution-free neural VLB estimators such as MINE, NWJ, CPC, or JS-based bounds, which operate directly on samples without imposing analytic distributional forms. We therefore view VNMC and VPCE as complementary approaches that target MI estimation under different assumptions, rather than methods intended for the same training setting considered in this paper.

**Two-Stage Estimators.** Several other neural MI estimators in the literature use a two-stage design, where a separate model is trained to estimate MI before it is used in an optimization task, see for example, Letizia et al. (2024); Shalev et al. (2022); Liao et al. (2020); Tsai et al. (2020); Song & Ermon (2019). These approaches can provide stable estimates, but they must be retrained whenever the joint distribution changes, which makes them impractical for learning settings where the data features change during training. These methods operate differently from single-stage estimators and cannot be used directly during training.

**Applying Standard VLBs to Improve GANs.** All the estimators discussed are often used within learning problems where MI guides the optimization. One example is to examine their behavior within the GAN family, which provides a natural setting for incorporating MI estimation into a downstream task. Models in this family, including GANs and VAE-GANs (Goodfellow et al., 2014; Larsen et al., 2016), use a generator to create samples, a discriminator to judge their quality, and in VAE-GANs an encoder that maps data to latent codes. MI has been introduced into these models to better align latent variables with generated outputs. InfoGAN (Chen et al., 2016) encouraged part of the latent code to remain informative about the output by adding an MI term to the objective. Building on this idea, Belghazi et al. (2018) used MINE to increase the MI between data and their encoded representations in a BiGAN-style model (Donahue et al., 2016; Dumoulin et al., 2016), improving data coverage and reducing mode collapse. These examples show that MI objectives can help GAN variants learn richer representations and produce more diverse samples.

## 3  Methodology

In this section, we present the main contribution of this work, a Bayesian bootstrap resampling for MINE. We begin with a brief background on BNPL and then present the resulting bootstrap algorithms. Finally, we illustrate how this framework can be applied in a downstream task.

### 3.1  Dirichlet Process for the Modeling Data Distribution

The DP is an infinite generalization of the Dirichlet distribution that is considered on the sample space denoted as $\mathfrak{X}$, which possesses a $\sigma$-algebra $\mathcal{A}$ comprising subsets of $\mathfrak{X}$ (Ferguson, 1973). $F$ follows a DP with parameters $(a, H)$ with the notation $F \sim DP(a, H)$, if for any measurable partition $A_1, \ldots, A_k$ of $\mathfrak{X}$ with $k \geq 2$, the joint distribution of the vector $(F(A_1), \ldots, F(A_k))$ follows a Dirichlet distribution characterized by parameters $(aH(A_1), \ldots, aH(A_k))$. Moreover, it is assumed that $H(A_j) = 0$ implies $F(A_j) = 0$ with probability one. The base measure $H$ captures the prior knowledge regarding the data distribution, while $a$ signifies the strength or intensity of this knowledge.

As a conjugate prior, the Dirichlet process yields a posterior that is also a DP. Specifically, for $n$ independent and identically distributed draws ($\mathbf{X}_{1:n} \in \mathbb{R}^d$) from the random probability measure $F$, the posterior distribution of $F$ is

$$F^{\mathrm{Pos}} := (F \mid \mathbf{X}_{1:n}) \sim \mathrm{DP}(a + n, H_n^*). \tag{5}$$

The updated base measure is

$$H_n^* = \frac{a}{a + n} H + \frac{n}{a + n} F_{\mathbf{X}_{1:n}}, \tag{6}$$

where $F_{\mathbf{X}_{1:n}}$ denotes the empirical cumulative distribution function of the sample $\mathbf{X}_{1:n}$.

It can be shown that a Dirichlet process has an infinite series representation and is therefore a discrete probability measure. Although the stick-breaking representation is a commonly employed series representation for DP inference (Sethuraman, 1994), it lacks the necessary normalization terms to convert it into a probability measure (Zarepour & Al-Labadi, 2012). Additionally, simulating from an infinite series is only feasible through using a random truncation approach to handle the terms within the series. To address these limitations, Ishwaran & Zarepour (2002) introduced an approximation of the DP in the form of a finite series, which allows for convenient simulation. In the context of posterior inference, this approximation is given by

$$F_N^{\mathrm{Pos}} := \sum_{i=1}^{N} J_i^{\mathrm{Pos}} \delta_{\mathbf{X}_i^{\mathrm{Pos}}}, \tag{7}$$

where $\left(J_{1:N}^{\mathrm{Pos}}\right) \sim \mathrm{Dirichlet}((a+n)/N, \ldots, (a+n)/N)$, $\left(\mathbf{X}_{1:N}^{\mathrm{Pos}}\right) \overset{\mathrm{iid}}{\sim} H_n^*$, and $\delta_{\mathbf{X}^{\mathrm{Pos}}}$ is the Dirac delta measure. In this study, the variables $J_i^{\mathrm{Pos}}$ and $\mathbf{X}_i^{\mathrm{Pos}}$ represent the DP's weight and location, respectively. The sequence $(F_N^{\mathrm{Pos}})_{N \geq 1}$ converges in distribution to $F^{\mathrm{Pos}}$, where $F_N^{\mathrm{Pos}}$ and $F^{\mathrm{Pos}}$ are random values in $M_1(\mathbb{R}^d)$, the space of probability measures on $\mathbb{R}^d$ endowed with the topology of weak convergence (Ishwaran & Zarepour, 2002).

Since the approximation in equation 7 is finite by construction, the remaining practical question concerns the choice of the truncation level $N$. Rather than fixing $N$ in advance, an adaptive truncation rule is employed to determine the number of retained atoms so that additional terms contribute negligibly to the total mass. Following Zarepour & Al-Labadi (2012), let $V_i \overset{\mathrm{iid}}{\sim} \mathrm{Gamma}\left(\frac{a+n}{j}, 1\right)$ for $i = 1, \ldots, j$, and define the truncation level as

$$N = \inf \left\{ j : J_j^{\mathrm{pos}} = \frac{V_j}{\sum_{i=1}^{j} V_i} < \epsilon \right\}, \tag{8}$$

for a prescribed threshold $\epsilon \in (0, 1)$. This adaptive selection of $N$ ensures that the finite approximation $F_N^{\mathrm{Pos}}$ captures the dominant posterior mass while remaining computationally tractable, and it is used throughout the remainder of the paper.

### 3.2 BNPL: Posterior Bootstrap Framework

Let $(\mathbf{X}_{1:n})$ be IID random vectors in $\mathbb{R}^d$ with common distribution $F \in \mathcal{P}(\mathbb{R}^d)$, where $\mathcal{P}(\mathbb{R}^d)$ denotes the space of Borel probability measures on $\mathbb{R}^d$. Suppose we are interested in a parameter $\boldsymbol{\psi} \in \Psi \subseteq \mathbb{R}^p$ indexing a collection of parametric models

$$\mathcal{F}_\Psi = \{\, F_{\boldsymbol{\psi}}(\mathbf{x}) : \boldsymbol{\psi} \in \Psi \,\}, \tag{9}$$

for example, the distribution induced by a neural network family parameterized by $\boldsymbol{\psi}$.

Letting $F \notin \mathcal{F}_\Psi$, which allows for the possibility of model misspecification, the nonparametric learning (NPL) framework defines the parameter of interest as a functional $\dot{\boldsymbol{\psi}} : \mathcal{P}(\mathbb{R}^d) \to \Psi$ of the data distribution, given by

$$\dot{\boldsymbol{\psi}}(F) = \arg\min_\Psi \int L(\mathbf{x}, \boldsymbol{\psi})\, dF(\mathbf{x}), \tag{10}$$

where $L(\mathbf{x}, \boldsymbol{\psi})$ is a user-chosen loss that determines the target statistic (Fong et al., 2019; Lyddon et al., 2018; 2019).

Since $F$ is unknown, it is modeled as a random probability measure with a DP prior $F \sim \mathrm{DP}(\alpha, H)$. The core idea of BNPL is that the posterior random measure $F^{\mathrm{Pos}}$ induces a posterior distribution on the parameter of interest through the functional $\hat{\boldsymbol{\psi}} : \mathcal{P}(\mathbb{R}^d) \to \Psi$, given by

$$\hat{\boldsymbol{\psi}}(F^{\mathrm{Pos}}) = \arg\min_\Psi \int L(\mathbf{x}, \boldsymbol{\psi})\, dF^{\mathrm{Pos}}(\mathbf{x}). \tag{11}$$

Because the DP is discrete and we use the finite approximation in equation 7, the expression in equation 11 admits the practical approximation

$$\hat{\boldsymbol{\psi}}(F^{\mathrm{Pos}}) \approx \arg\min_\Psi \sum_{i=1}^N J_i^{\mathrm{Pos}}\, L(\mathbf{x}_i^{\mathrm{Pos}}, \boldsymbol{\psi}), \tag{12}$$

which corresponds to the Bayesian bootstrap procedure described in Fong et al. (2019, Algorithm 2), with the difference that Fong et al. (2019) employs a truncation of the stick-breaking series representation to obtain this finite approximation.

Dellaporta et al. (2022) and Fazeli-Asl et al. (2024) also independently introduced posterior bootstrap methods based on two different DP approximations for learning the data distribution. In both works, learning is carried out by minimizing the maximum mean discrepancy between the posterior distribution $F^{\mathrm{Pos}}$ and the distribution induced by a generator modeled as a parameterized neural network.

Overall, these posterior bootstrap methods operate within a robust learning framework and are designed to handle model misspecification.

### 3.3 DPMINE: Posterior Bootstrap Framework for MINE

We now present our main contribution by extending the Bayesian bootstrap idea of Fong et al. (2019) to the MINE. Although our setting differs from Fong et al. (2019), the connection is direct: the posterior on the MINE parameters is induced by placing a DP posterior on the unknown data distribution. This construction makes the original MINE applicable for MI-based BNPL. We first develop this approach for the KL-MINE objective and then show that the same steps apply to the JS-MINE case.

**KL-based representation:** Let $(\mathbf{X}_{1:n})$ be $n$ IID random variables with $\mathbf{X} \in \mathbb{R}^d \sim F$. For any continuous mappings $f_i : \mathbb{R}^d \to \mathbb{R}^{d'}$, $i = 1, 2$, we are interested in estimating the MI between $f_1(\mathbf{X})$ and $f_2(\mathbf{X})$ using the KL-MINE objective from the BNPL perspective. In this setting, we define the parameter of interest as the functional $\dot{\boldsymbol{\gamma}} : \mathcal{P}(\mathbb{R}^d) \to \boldsymbol{\Gamma}$, given by

$$\dot{\boldsymbol{\gamma}}(F) = \arg\min_{\boldsymbol{\gamma} \in \boldsymbol{\Gamma}} \big(-L_{\boldsymbol{\gamma}}^{\mathrm{KL}}(f_1(\mathbf{X}), f_2(\mathbf{X}))\big). \tag{13}$$

---

**Algorithm 1:** MINE Posterior Bootstrap

---

**Input:** training data $\mathbf{X}_{1:n}$, mappings $f_1, f_2$, concentration parameter $a$, base measure $H$, truncation level $N$, number of bootstrap samples $B$, critic network $T_{\boldsymbol{\gamma}}$

**1 for** $i = 1$ **to** $B$ **do**

**2** $\quad$ Select the $i$-th mini-batch $\mathbf{X}_{\mathrm{mb}}^{(i)}$ of size $n_{\mathrm{mb}}$;

**3** $\quad$ Define $H_{n_{\mathrm{mb}}}^* = \frac{a}{a+n_{\mathrm{mb}}} H + \frac{n_{\mathrm{mb}}}{a+n_{\mathrm{mb}}} F_{\mathbf{X}_{\mathrm{mb}}^{(i)}}$;

**4** $\quad$ Draw posterior samples $\mathbf{X}_{1:N}^{\mathrm{Pos}} \overset{\text{iid}}{\sim} H_{n_{\mathrm{mb}}}^*$;

**5** $\quad$ Draw Dirichlet weights $(J_{1:N}^{\mathrm{Pos}}) \sim \mathrm{Dirichlet}((a+n)/N, \ldots, (a+n)/N)$;

**6** $\quad$ Compute $L_{\boldsymbol{\gamma}}^{\mathrm{DP}}(\cdot, \cdot)$ using equation 15 (DPKL) or equation 17 (DPJS) with the critic $T_{\boldsymbol{\gamma}}$, posterior samples $\mathbf{X}_{1:N}^{\mathrm{Pos}}$, and Dirichlet weights $J_{1:N}^{\mathrm{Pos}}$;

**7** $\quad$ Update the critic parameter

$$\hat{\boldsymbol{\gamma}}^{(i)} = \arg\min_{\boldsymbol{\gamma} \in \boldsymbol{\Gamma}} \Big\{ -L_{\boldsymbol{\gamma}}^{\mathrm{DP}}\big(f_1(\mathbf{X}_{1:N}^{\mathrm{Pos}}), f_2(\mathbf{X}_{1:N}^{\mathrm{Pos}})\big) \Big\}$$

**8** $\quad$ Compute the MI estimate

$$\mathrm{MI}^{\mathrm{DP}(i)} = L_{\hat{\boldsymbol{\gamma}}^{(i)}}^{\mathrm{DP}}\big(f_1(\mathbf{X}_{1:N}^{\mathrm{Pos}}), f_2(\mathbf{X}_{1:N}^{\mathrm{Pos}})\big)$$

**9 return** Posterior bootstrap sample $\mathrm{MI}^{\mathrm{DP}(1:B)}$

---

Given a posterior draw $F^{\mathrm{Pos}} \sim \mathrm{DP}(a + n, H_n^*)$, the posterior for $\dot{\boldsymbol{\gamma}}(F)$ is obtained through the functional $\hat{\boldsymbol{\gamma}} : \mathcal{P}(\mathbb{R}^d) \to \boldsymbol{\Gamma}$, defined by

$$\hat{\boldsymbol{\gamma}}(F^{\mathrm{Pos}}) = \arg\min_{\boldsymbol{\gamma} \in \boldsymbol{\Gamma}} \big\{ -L_{\boldsymbol{\gamma}}^{\mathrm{DPKL}}(f_1(\mathbf{X}_{1:N}^{\mathrm{Pos}}), f_2(\mathbf{X}_{1:N}^{\mathrm{Pos}})) \big\}, \tag{14}$$

where

$$L_{\boldsymbol{\gamma}}^{\mathrm{DPKL}}(f_1(\mathbf{X}_{1:N}^{\mathrm{Pos}}), f_2(\mathbf{X}_{1:N}^{\mathrm{Pos}})) := \sum_{\ell=1}^{N} J_{\ell}^{\mathrm{Pos}} T_{\boldsymbol{\gamma}}(f_1(\mathbf{X}_{\ell}^{\mathrm{Pos}}), f_2(\mathbf{X}_{\ell}^{\mathrm{Pos}})) - \ln \sum_{\ell=1}^{N} J_{\ell}^{\mathrm{Pos}} e^{T_{\boldsymbol{\gamma}}(f_1(\mathbf{X}_{\ell}^{\mathrm{Pos}}), f_2(\mathbf{X}_{\pi(\ell)}^{\mathrm{Pos}}))}. \tag{15}$$

Finally, equation 16 represents the posterior belief about $\mathrm{MI}^{\mathrm{KL}}(f_1(\mathbf{X}), f_2(\mathbf{X}))$.

$$\mathrm{MI}^{\mathrm{DPKL}}\big(f_1(\mathbf{X}^{\mathrm{Pos}}), f_2(\mathbf{X}^{\mathrm{Pos}})\big) = L_{\hat{\boldsymbol{\gamma}}}^{\mathrm{DPKL}}\big(f_1(\mathbf{X}^{\mathrm{Pos}}), f_2(\mathbf{X}^{\mathrm{Pos}})\big). \tag{16}$$

Algorithm 1 provides a clear illustration of these steps.

**JS-based representation:** All steps used in the Bayesian bootstrap construction for KL-MINE extend simply to the JS-MINE objective. The DP-based analogue of the JS variational bound is given by

$$L_{\boldsymbol{\gamma}}^{\mathrm{DPJS}}(f_1(\mathbf{X}_{1:N}^{\mathrm{Pos}}), f_2(\mathbf{X}_{1:N}^{\mathrm{Pos}})) := \sum_{\ell=1}^{N} J_{\ell}^{\mathrm{Pos}} \Big[ -\zeta(-T_{\boldsymbol{\gamma}}(f_1(\mathbf{X}_{\ell}^{\mathrm{Pos}}), f_2(\mathbf{X}_{\ell}^{\mathrm{Pos}}))) - \zeta(T_{\boldsymbol{\gamma}}(f_1(\mathbf{X}_{\ell}^{\mathrm{Pos}}), f_2(\mathbf{X}_{\pi(\ell)}^{\mathrm{Pos}}))) \Big], \tag{17}$$

The following theorem states that the proposed Bayesian VLB estimator is asymptotically no smaller, on average, than the standard VLB used in MINE. This results in a tighter VLB on the true MI and allows for a more accurate estimation of MI by maximizing the tighter bound over a compact domain $\boldsymbol{\Gamma}$.

**Theorem 1** (Limiting expectation). *Considering DP posterior representations defined in equation 15 and equation 17. Let both $T_{\boldsymbol{\gamma}}$ and $e^{T_{\boldsymbol{\gamma}}}$ are $M$-bounded (i.e., $|T_{\boldsymbol{\gamma}}|, |e^{T_{\boldsymbol{\gamma}}}| \leq M$). Given the DP posterior approximation in equation 7, for any $N > 0$, we have,*

$$i. \ \lim_{n \to \infty} \mathbb{E}_{F_N^{Pos}} \left( L_{\hat{\boldsymbol{\gamma}}}^{DPKL}(f_1(\mathbf{X}_{1:N}^{Pos}), f_2(\mathbf{X}_{1:N}^{Pos})) \right) \geq L_{\boldsymbol{\gamma}}^{KL}(f_1(\mathbf{X}), f_2(\mathbf{X})),$$

$ii.$ $\lim\limits_{n\to\infty} \mathbb{E}_{F_N^{Pos}} \left( L_{\hat{\boldsymbol{\gamma}}}^{DPJS}(f_1(\mathbf{X}_{1:N}^{Pos}), f_2(\mathbf{X}_{1:N}^{Pos})) \right) \geq L_{\boldsymbol{\gamma}}^{JS}(f_1(\mathbf{X}), f_2(\mathbf{X})).$

In Theorem 1, the expectation is taken with respect to the nonparametric posterior on the data-generating process. As a result, the randomness in $\hat{\boldsymbol{\gamma}}$, which arises from the posterior bootstrap, is automatically averaged out. Equivalently, $\hat{\boldsymbol{\gamma}}$ is treated as fixed conditional on a posterior draw, and the theorem characterises the resulting average behaviour. This treatment is standard in Bayesian nonparametric learning and posterior bootstrap analyses (Dellaporta et al., 2022).

### 3.4 Application: Integrating DPMINE into VAE-GAN Models within BNPL

The BNPWMMD-GAN framework proposed by Fazeli-Asl & Zhang (2023) leverages representation learning within BNPL by placing a DP prior $F \sim \mathrm{DP}(a, H)$ to generate synthetic samples.

The model follows a standard VAE-GAN architecture, consisting of an encoder $E_{\boldsymbol{\eta}} : \mathbb{R}^d \to \mathbb{R}^p$, $\boldsymbol{\eta} \in \mathcal{H}$, that maps data into a latent representation, a generator (decoder) $G_{\boldsymbol{\omega}} : \mathbb{R}^p \to \mathbb{R}^d$, $\boldsymbol{\omega} \in \boldsymbol{\Omega}$, that reconstructs and generates samples from latent codes, a code generator $CG_{\boldsymbol{\omega}'} : \mathbb{R}^q \to \mathbb{R}^p$, $\boldsymbol{\omega}' \in \boldsymbol{\Omega}'$, with $q < p$ that injects additional latent noise to improve coverage of underrepresented regions of the latent space, and a discriminator $D_{\boldsymbol{\theta}} : \mathbb{R}^d \to \mathbb{R}$, $\boldsymbol{\theta} \in \boldsymbol{\Theta}$, that distinguishes real from generated samples. These components are trained jointly using Wasserstein and MMD-based losses evaluated on posterior-weighted samples generated by DP approximation equation 7. Further details are provided in Appendix C.

Training is implemented by estimating the network parameters as functionals of $F^{\mathrm{Pos}}$. Let

$$\boldsymbol{\Xi} = \boldsymbol{\Omega} \times \mathcal{H} \times \boldsymbol{\Theta} \times \boldsymbol{\Omega}'$$

denote the joint parameter space. The baseline BNPWMMD-GAN objective is defined as

$$(\hat{\boldsymbol{\omega}}, \hat{\boldsymbol{\eta}}, \hat{\boldsymbol{\theta}}, \hat{\boldsymbol{\omega}}')(F^{\mathrm{Pos}}) = \arg \min_{(\boldsymbol{\omega}, \boldsymbol{\eta}, \boldsymbol{\theta}, \boldsymbol{\omega}') \in \boldsymbol{\Xi}} \left\{ L(G_{\boldsymbol{\omega}}, E_{\boldsymbol{\eta}}, F_N^{\mathrm{Pos}}) + L(D_{\boldsymbol{\theta}}, F_N^{\mathrm{Pos}}) + L(CG_{\boldsymbol{\omega}'}, F_N^{\mathrm{Pos}}) \right\}. \tag{18}$$

To improve information extraction during encoding and preserve meaningful structure during decoding, this objective is refined by two regularization terms estimated via the proposed DPMINE framework, yielding

$$(\hat{\boldsymbol{\omega}}, \hat{\boldsymbol{\eta}}, \hat{\boldsymbol{\theta}}, \hat{\boldsymbol{\omega}}')(F^{\mathrm{Pos}}) = \arg \min_{(\boldsymbol{\omega}, \boldsymbol{\eta}, \boldsymbol{\theta}, \boldsymbol{\omega}') \in \boldsymbol{\Xi}} \Big\{ L(G_{\boldsymbol{\omega}}, E_{\boldsymbol{\eta}}, F_N^{\mathrm{Pos}}) + L(D_{\boldsymbol{\theta}}, F_N^{\mathrm{Pos}}) + L(CG_{\boldsymbol{\omega}'}, F_N^{\mathrm{Pos}})$$

$$- L_{\hat{\boldsymbol{\gamma}}_1}^{\mathrm{DP}} \left( \mathbf{X}_{1:N}^{\mathrm{Pos}}, E_{\boldsymbol{\eta}}(\mathbf{X}_{1:N}^{\mathrm{Pos}}) \right) \tag{19}$$

$$- L_{\hat{\boldsymbol{\gamma}}_2}^{\mathrm{DP}} \left( E_{\boldsymbol{\eta}}(\mathbf{X}_{1:N}^{\mathrm{Pos}}), G_{\boldsymbol{\omega}}(E_{\boldsymbol{\eta}}(\mathbf{X}_{1:N}^{\mathrm{Pos}})) \right) \Big\}. \tag{20}$$

The regularization terms in equation 19 and equation 20, computed using the corresponding critics $T_{\hat{\boldsymbol{\gamma}}_1}$ and $T_{\hat{\boldsymbol{\gamma}}_2}$, estimate $\mathrm{MI}(X, E_{\boldsymbol{\eta}}(X))$ and $\mathrm{MI}(E_{\boldsymbol{\eta}}(X), G_{\boldsymbol{\omega}}(E_{\boldsymbol{\eta}}(X)))$, respectively. The associated critic parameters are estimated as posterior functionals,

$$\hat{\boldsymbol{\gamma}}_1(F^{\mathrm{Pos}}) = \arg \min_{\boldsymbol{\gamma}_1 \in \boldsymbol{\Gamma}_1} \left\{ -L_{\boldsymbol{\gamma}_1}^{\mathrm{DP}} \left( \mathbf{X}_{1:N}^{\mathrm{Pos}}, E_{\boldsymbol{\eta}}(\mathbf{X}_{1:N}^{\mathrm{Pos}}) \right) \right\}, \tag{21}$$

$$\hat{\boldsymbol{\gamma}}_2(F^{\mathrm{Pos}}) = \arg \min_{\boldsymbol{\gamma}_2 \in \boldsymbol{\Gamma}_2} \left\{ -L_{\boldsymbol{\gamma}_2}^{\mathrm{DP}} \left( E_{\boldsymbol{\eta}}(\mathbf{X}_{1:N}^{\mathrm{Pos}}), G_{\boldsymbol{\omega}}(E_{\boldsymbol{\eta}}(\mathbf{X}_{1:N}^{\mathrm{Pos}})) \right) \right\}. \tag{22}$$

Overall, this formulation integrates DPMINE terms into a VAE-GAN training procedure, while maintaining the BNPL interpretation.

Using the notation of Section 3.3, the term in equation 21 corresponds to

$$f_1 \left( \mathbf{X}_{1:N}^{\mathrm{Pos}} \right) = \mathbf{X}_{1:N}^{\mathrm{Pos}}, \qquad f_2 \left( \mathbf{X}_{1:N}^{\mathrm{Pos}} \right) = E_{\boldsymbol{\eta}} \left( \mathbf{X}_{1:N}^{\mathrm{Pos}} \right), \tag{23}$$

Similarly, the term in equation 22 corresponds to

$$f_1 \left( \mathbf{X}_{1:N}^{\mathrm{Pos}} \right) = E_{\boldsymbol{\eta}} \left( \mathbf{X}_{1:N}^{\mathrm{Pos}} \right), \qquad f_2 \left( \mathbf{X}_{1:N}^{\mathrm{Pos}} \right) = G_{\boldsymbol{\omega}} \left( E_{\boldsymbol{\eta}} \left( \mathbf{X}_{1:N}^{\mathrm{Pos}} \right) \right), \tag{24}$$

when $G_{\boldsymbol{\omega}}$ acts as the decoder.

# 4 Experimental Results

In this section, the procedure is evaluated on both simulated and real datasets. All experiments are conducted using mini-batches, with the batch size chosen appropriately for each setting. To place a DP prior on the data distribution $F$, its hyperparameters must first be specified. Unless an experiment is designed to study their specific influence, the base measure is taken to be a multivariate normal distribution estimated from each mini-batch. For a mini-batch $\mathbf{X}_{\mathrm{mb}}^{(i)}$, $i \in \{1, \ldots, B\}$, the sample mean and covariance are

$$\overline{\mathbf{X}}_{\mathrm{mb}} = \frac{1}{B} \sum_{i=1}^{n} \mathbf{X}_{\mathrm{mb}}^{(i)}, \quad S_{\mathbf{X}_{\mathrm{mb}}} = \sum_{i=1}^{B} (\mathbf{X}_{\mathrm{mb}}^{(i)} - \overline{\mathbf{X}}_{\mathrm{mb}})(\mathbf{X}_{\mathrm{mb}}^{(i)} - \overline{\mathbf{X}}_{\mathrm{mb}})^{T}, \tag{25}$$

and the base measure is defined as

$$H := N(\overline{\mathbf{X}}_{\mathrm{mb}}, S_{\mathbf{X}_{\mathrm{mb}}}).$$

The concentration parameter is then selected as the *maximum a posteriori* (MAP) estimate of $a$, obtained by maximizing the log-likelihood of $F^{\mathrm{Pos}}$ with respect to $a$. Full details of this computation are provided in Fazeli-Asl & Zhang (2023, Section 4.2). Additionally, the value of $N$ in the DP approximation of equation 7 is chosen as prescribed in Section 3.1, using $\epsilon = 10^{-4}$.

We start by presenting simulation studies that evaluate the KL-based and JS-based versions of our Bayesian bootstrap framework for the MI estimation. We then show how the better-performing estimator can be used to refine the training of VAE-GANs within the BNPL framework.

## 4.1 MI Estimation using Bayesian Bootstrap Resampling

We follow the staircase protocol of Letizia et al. (2024), adapted to our experimental setting, to construct the toy examples. In our design, we let

$$\mathbf{X} = (\mathbf{U}, \mathbf{V}) \in \mathbb{R}^{2k}, \qquad \mathbf{U}, \mathbf{V} \in \mathbb{R}^{k},$$

and consider estimating $\mathrm{MI}(\mathbf{U}, \mathbf{V})$ for $k \in \{1, 2, 3, 4, 5\}$. Increasing $k$ produces the successive staircase levels: the first stage uses $k = 1$, the second uses $k = 2$, and this continues until the fifth stage uses $k = 5$. This yields five distinct increases in the true MI, giving a clear benchmark for comparing estimator performance as more dimensions are added.

**Binary-Gaussian Example.** For each value of $k$, we generate a binary-Gaussian example by defining

$$\mathbf{U} = \mathrm{sign}(\mathbf{Z}), \qquad \mathbf{Z} \sim N(\mathbf{0}, \mathbf{I}_k), \qquad \mathbf{V} = \mathbf{U} + \boldsymbol{\epsilon}, \qquad \boldsymbol{\epsilon} \sim N(\mathbf{0}, 0.2\,\mathbf{I}_k).$$

In the notation of Section 3.3, where the MI was written as $\mathrm{MI}(f_1(\mathbf{X}), f_2(\mathbf{X}))$, this setting corresponds to the choice $f_1(\mathbf{X}) = \mathbf{U}$ and $f_2(\mathbf{X}) = \mathbf{V}$.

Finally, each stairstep provides 4000 training steps, where a step means one gradient update of the neural estimation tasks. At every step, we draw a mini-batch from the distribution associated with the current value of $k$. Additional toy examples following the same protocol are provided in the appendix.

### 4.1.1 DPMINE versus MINE

As the main contribution of this paper is to apply Bayesian bootstrap resampling to the mini-batches used in MINE, we begin by examining whether our Bayesian version can outperform the original estimator. Figure 1 shows our results, where we use a mini-batch size of 16. The solid curves display the running averages of each estimator; for BNP-MINE, this running average corresponds to the posterior expectation induced by the Bayesian bootstrap and represents the expected behavior of the MI posterior, which converges to the true MI in the KL-based setting. The shaded regions show the raw mini-batch variability. In the KL setting, our BNP-MINE estimator gives a tighter and more stable lower bound compared to FNP-MINE, which numerically supports the theoretical results in Theorem 1(i). The FNP curves vary widely and can rise

above the true MI, reflecting the instability often seen in MINE training. In contrast, the BNP-KL curve has less variance and remains close to the true MI, leading to a more consistent estimate.

The BNP-JS also shows reduced variance and avoids the large jumps observed in the FNP-JS, even though it is not as tight as the KL version. Overall, the BNP framework leads to more stable and reliable MI estimates, with the KL formulation giving the strongest improvement.

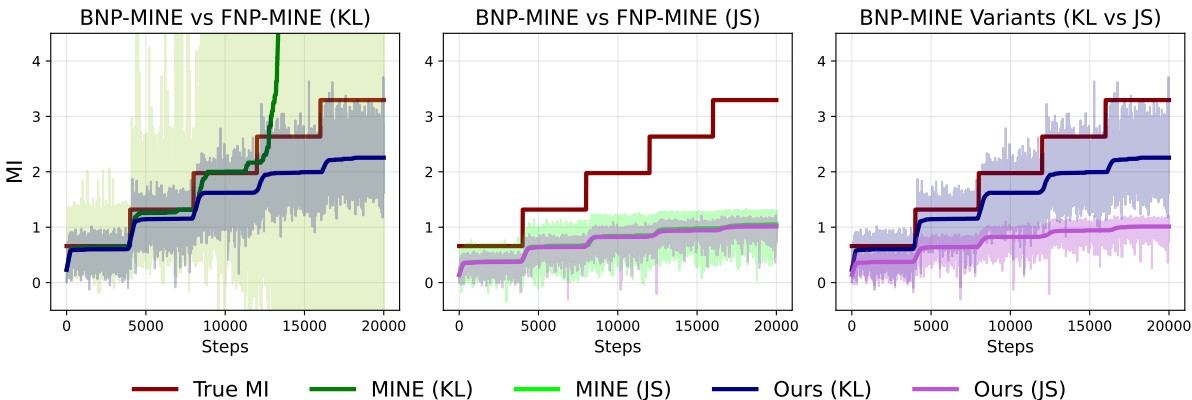

Figure 1: Staircase scheme for the binary-Gaussian setting with $\mathbf{X} = (\mathbf{U}, \mathbf{V}) \sim F$. BNP-MINE and FNP-MINE (KL and JS) are compared using mini-batch size 16, with $F \sim \mathrm{DP}(a_{\mathrm{MAP}}, H)$ and $H := N(\overline{\mathbf{X}}_{\mathrm{mb}}, S_{\mathbf{X}_{\mathrm{mb}}})$, across dimensions $k = 1, \ldots, 5$.

### 4.1.2 Sensitivity to different mini-batch sizes

To examine how different mini-batch sizes affect the estimator, Figure 2 shows BNP-MINE under several choices of batch size. Across all settings, the estimates remain close to the true MI, and larger batches tend to produce smoother and slightly more accurate curves. Overall, the method remains stable, and the effect of the batch size is modest.

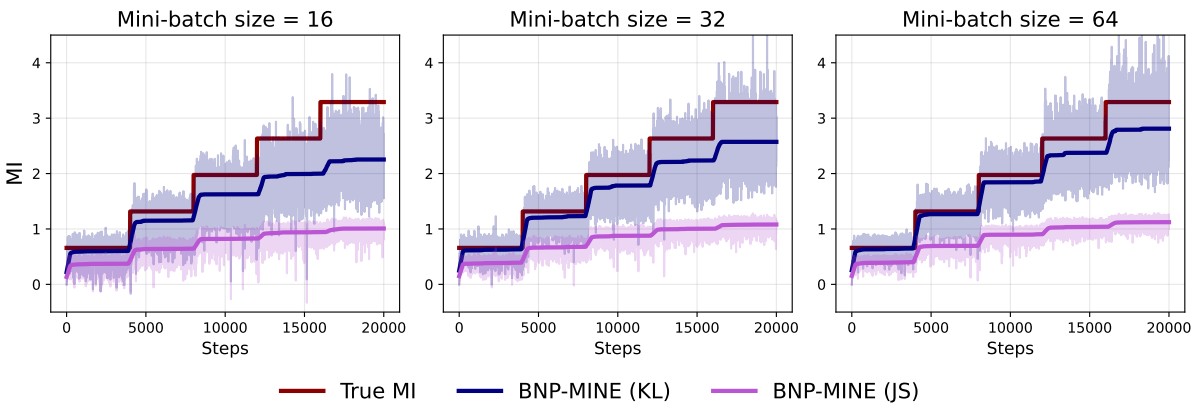

Figure 2: Staircase scheme for the binary-Gaussian setting with $\mathbf{X} = (\mathbf{U}, \mathbf{V}) \sim F$. BNP-MINE is evaluated across different mini-batch sizes, with $F \sim \mathrm{DP}(a_{\mathrm{MAP}}, H)$ and $H := N(\overline{\mathbf{X}}_{\mathrm{mb}}, S_{\mathbf{X}_{\mathrm{mb}}})$, across dimensions $k = 1, \ldots, 5$.

### 4.1.3 Sensitivity to different choices of DP hyperparameters

To examine how the DP hyperparameters affect our procedure, we study a range of concentration values $a \in \{0.005, 25, 250, 2500\}$ as well as $a_{\mathrm{MAP}}$, under two choices of the centering measure $H$. In both sets of experiments, $a_{\mathrm{MAP}}$ produces stable estimates in the sense that the MI remains close to the true value.

Figure 3(a) shows that when $H$ is taken to be a standard normal distribution, increasing $a$ forces the prior to dominate the data, and the estimator gradually converges to the MI of a standard normal model, which is zero. In contrast, Figure 3(b) shows that when $H$ is defined using the sample mean and sample covariance of each mini-batch, the prior remains aligned with the data, and the estimator stays accurate even for large $a$. These results, which follow directly from the posterior base measure in equation 6, show that reasonable choices of $H$ keep the method reliable even for large $a$. Although strongly misspecified choices of $H$ can push the estimates toward the wrong target, the method remains robust to the choice of $H$ when $a$ is sufficiently small.

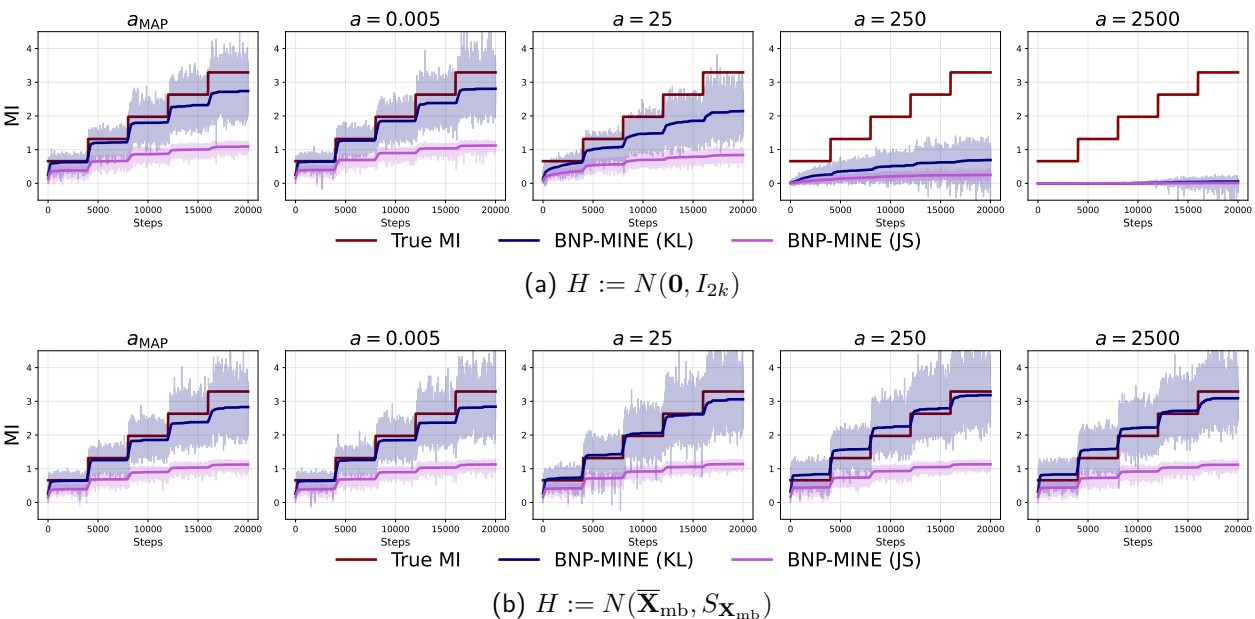

(a) $H := N(\mathbf{0}, I_{2k})$

(b) $H := N(\overline{\mathbf{X}}_{\mathrm{mb}}, S_{\mathbf{X}_{\mathrm{mb}}})$

Figure 3: Staircase scheme for the binary-Gaussian setting with $\mathbf{X} = (\mathbf{U}, \mathbf{V}) \sim F$. BNP-MINE is evaluated using a mini-batch size of 64, with $F \sim \mathrm{DP}(a, H)$ under different choices of the concentration parameter $a$ and the centering measure $H$, across dimensions $k = 1, \ldots, 5$.

### 4.1.4 Comparison with further differentiable MI-VLB estimators

In addition to the traditional MINE estimator (Belghazi et al., 2018), we compare our method against several influential differentiable MI-VLB approaches. NWJ (Nguyen et al., 2010) is unbiased but typically yields a loose and high-variance lower bound. CPC/InfoNCE (Oord et al., 2018) offers stable optimization but is fundamentally constrained by $\ln(n_{\mathrm{mb}})$, restricting its ability to capture high MI. We also include JSD-LB (Dorent et al., 2025), the most recent state-of-the-art estimator providing a tight and tractable lower bound connecting JS divergence to MI. As our experiments show, BNP-MINE delivers a more stable and reliable lower-bound estimate than NWJ and JSD-based methods, and although CPC appears smoother, its structural upper bound prevents consistent performance across settings. Additional results in the appendix demonstrate that CPC becomes non-tight in several benchmarks, while our estimator consistently preserves the staircase structure with substantially reduced variance.

**Sensitivity to high dimensionality.** To examine how well our method handles high-dimensional data, we test it with $k = 10, 100, 1000$ as shown in Figure 5. In all cases, DPMINE produces smooth and stable curves with much lower noise compared to the baseline estimators. As the dimension increases, MINE and NWJ become noticeably more unstable, while DPMINE continues to give a steady estimate of the MI lower bound. These results show that our estimator remains reliable even in very high dimensional settings.

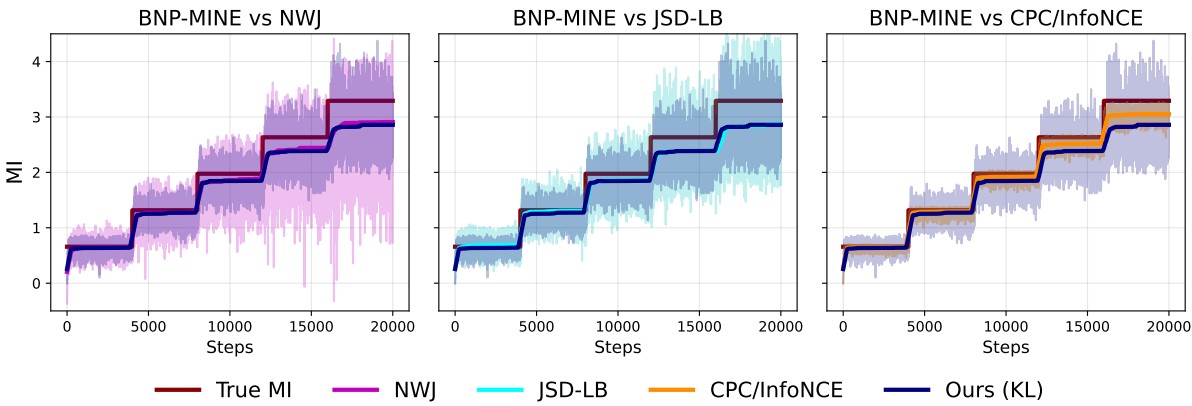

Figure 4: Staircase scheme for the binary-Gaussian setting with $\mathbf{X} = (\mathbf{U}, \mathbf{V}) \sim F$. BNP-MINE is compared against different baselines using a fixed mini-batch size of 64, with $F \sim \mathrm{DP}(a_{\mathrm{MAP}}, H)$ and $H := N(\overline{\mathbf{X}}_{\mathrm{mb}}, S_{\mathbf{X}_{\mathrm{mb}}})$, across dimensions $k = 1, \ldots, 5$.

Table 1: Runtime of neural variational lower-bound estimators across data dimensions and training epochs.

| Estimator | Epoch | Dimension ($d = 2 \times k$) | | | |
|---|---|---|---|---|---|
| | | **2** | **20** | **200** | **2000** |
| MINE, NWJ, | 500 | 5 sec | 5 sec | 5 sec | 5 sec |
| CPC/InfoNCE, JSD-LB | 1500 | 12 sec | 12 sec | 12 sec | 12 sec |
| DPMINE | 500 | 42 sec | 42 sec | 42 sec | 42 sec |
| | 1500 | 1 min 25 sec | 1 min 25 sec | 1 min 25 sec | 1 min 25 sec |

## 4.2 Improving VAE-GAN training using DPMINE within BNPL

To implement the BNPWMMD+DPMINE, we consider the Gaussian kernel function, defined as $k_{G_\sigma}(\mathbf{X}, \mathbf{Y}) = \exp(\frac{-||\mathbf{X}-\mathbf{Y}||^2}{2\sigma^2})$ with bandwidth parameter $\sigma$, in the MMD distance given in Appendix B. We search for the appropriate bandwidth parameter $\sigma$ over a fixed grid of values, $\sigma \in \{2, 5, 10, 20, 40, 80\}$. We then compute the mixture of Gaussian kernels, denoted as $k(\cdot, \cdot) = \sum_\sigma k_{G_\sigma}(\cdot, \cdot)$. This selection of kernel function and bandwidth has been shown to yield satisfactory performance in training MMD-based GANs, as mentioned in Li et al. (2015); Fazeli-Asl et al. (2024); Fazeli-Asl & Zhang (2023). Then, we consider next examples to investigate the proposed approach.

### 4.2.1 Toy Example

**Coil Dataset:** We first look at a toy example that showcases the effectiveness of the BN-PWMMD+DPMINE in mitigating mode collapse. In this example, we simulate 5000 samples in 3D space, denoted as $(X(t), Y(t), Z(t))$, where $X(t) = 6\cos t$, $Y(t) = 6\sin t$, and $Z(t) = t$, with $t$ ranging from $-2\pi$ to $4\pi$. We then normalize all datasets to a range between $-1$ and $1$. This normalization ensures compatibility with the hyperbolic tangent activation function, which is used in the generator's last layer in all compared models. We also used a latent dimension of 100 with a sub-latent dimension of 10 in this example. Additionally, we provide the results of the BNPWMMD to display the basic model's performance in covering data space in the absence of DPMINE.

Figure 6 illustrates the effect of incorporating DPMINE into the BNPWMMD-GAN framework. Compared to BNPWMMD alone, BNPWMMD+DPMINE produces generated samples that more closely follow the geometric structure of the real data, with reduced dispersion and improved alignment along the underlying manifold. This improvement is particularly evident in the random generation results, where DP–MINE helps preserve global structure and continuity.

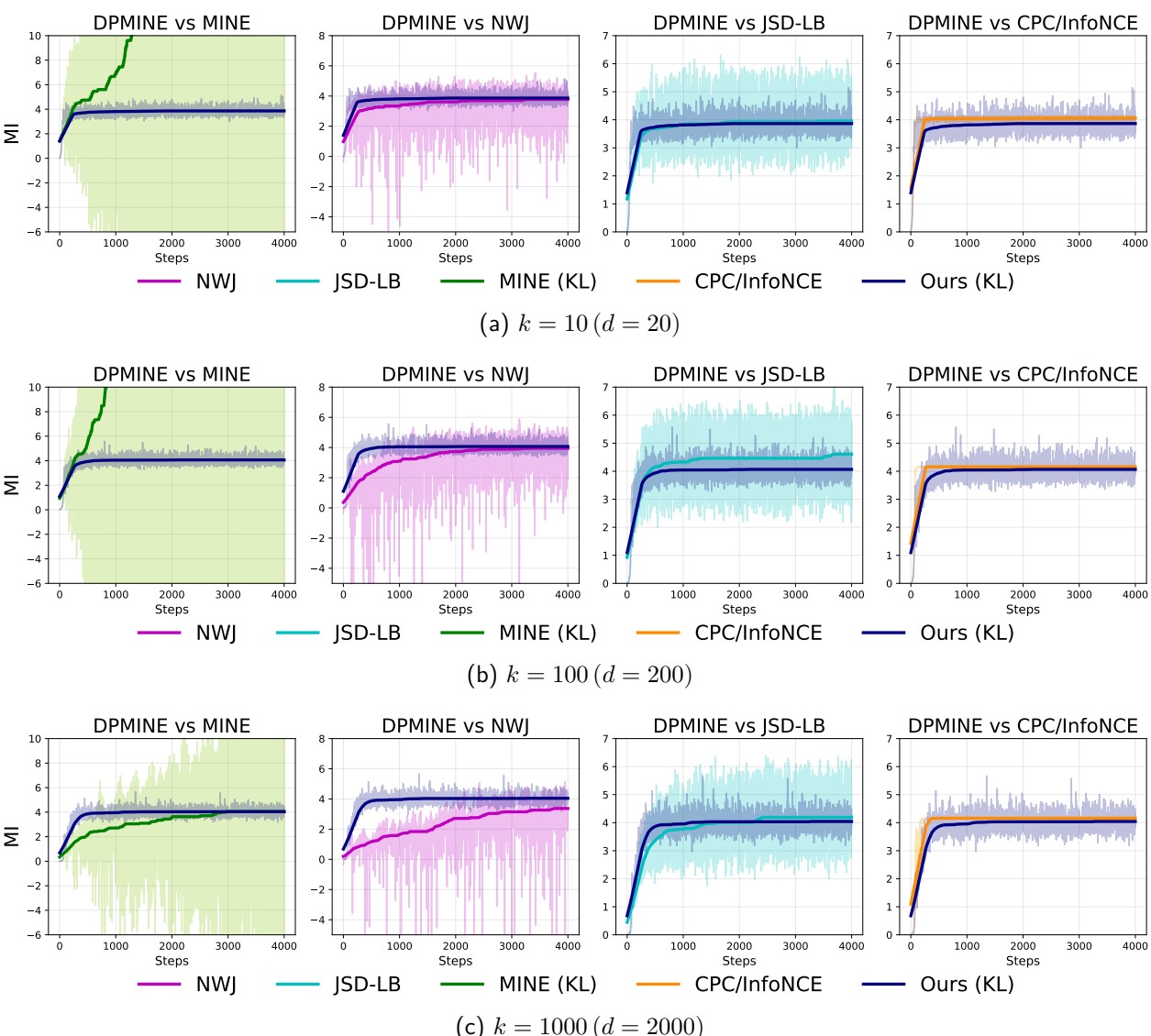

Figure 5: Binary-Gaussian setting with $\mathbf{X} = (\mathbf{U}, \mathbf{V}) \sim F$. BNP-MINE uses mini-batch size 64, with $F \sim \text{DP}(a_{\text{MAP}}, H)$ and $H = N(\overline{\mathbf{X}}_{\text{mb}}, S_{\mathbf{X}_{\text{mb}}})$. Experiments are run for $k = 10, 100, 1000$, using 4000 steps per dimension.

In the reconstruction task, the inclusion of DPMINE leads to more faithful reconstructions with smoother trajectories and fewer distortions, indicating improved information retention between the data and latent spaces. Overall, these results show that DP–MINE enhances representation quality, stabilizes training, and reduces mode collapse.

### 4.2.2 Real Example

**COVID-19 Dataset:** This dataset comprises of 3D chest CT images of 1000 subjects diagnosed with lung infections after testing positive for COVID-19[1]. All images in this dataset are stored in the DICOM format and have a resolution of 16 bits per pixel, with dimensions of $512 \times 512$ pixels in grayscale. We randomly selected 200 patients, resulting in a total of 91,960 images. As part of the preprocessing step, we first stored each patient's data in a Neuroimaging Informatics Technology Initiative (NIFTI) format file. Then, we

---

[1]The dataset is freely available online at `https://doi.org/10.7910/DVN/6ACUZJ` (license: CC0 1.0).

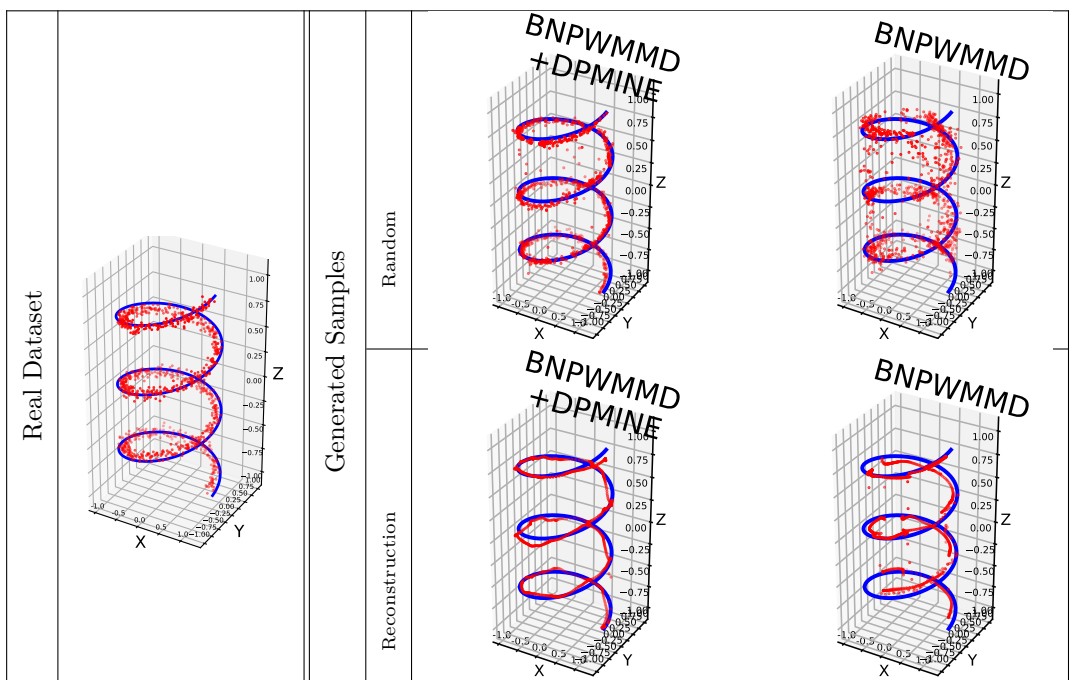

Figure 6: 1000 randomly generated and reconstructed samples using BNPWMMD+DPMINE, compared with BNPWMMD, after 5000 epochs for the coil example.

converted each NIFTI file into a 3D image with a dimension of $64 \times 64 \times 64$. Each dimension represents the axial, sagittal, and coronal views of the lungs. After normalizing the dataset, we used a mini-batch size of 16 and trained all compared models for 7500 epochs. We also used a latent dimension of $p = 1000$ as suggested in Kwon et al. (2019) for high-dimensional cases, with a sub-latent dimension of $q = 100$ as used in Fazeli-Asl & Zhang (2023).

To illustrate the image generation performance of BNPWMMD+DPMINE in comparison with BNPWMMD alone, Figure 7 presents a 3D visualization of real and randomly generated chest CT samples. The results indicate that BNPWMMD+DPMINE produces sharper and more coherent images. For a more comprehensive view of the generated samples, we included additional slices from each dimension representation in Appendix E.2.2. We also compared the reconstruction capability of different models by displaying slices of a reconstructed sample from each model in the Appendix. The BNPWMMD+DPMINE yields reconstructions that are visually closer to the training data under BNPWMMD+DPMINE.

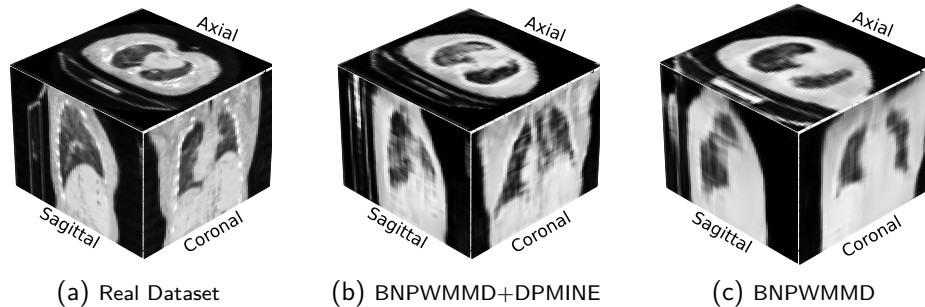

(a) Real Dataset          (b) BNPWMMD+DPMINE          (c) BNPWMMD

Figure 7: A 3D visualization of a real sample and a randomly generated sample using BN-PWMMD+DPMINE after 7500 epochs, compared with BNPWMMD, in the COVID-19 example.

However, additional quantitative tools are essential for assessing the similarity between real and generated samples. Dimensionality reduction provides a useful way to visualize how well generated samples align with the real data distribution.

In Figure 8(a), the t-SNE embedding (Van der Maaten & Hinton, 2008) of the real data and 200 randomly generated samples shows that BNPWMMD+DPMINE achieves broader and more consistent coverage of the real data manifold compared to BNPWMMD alone. In contrast, samples generated by BNP–WMMD appear more concentrated and exhibit gaps relative to the real data distribution.

Figure 8(b) provides a complementary view using features obtained from a custom two-layer linear encoder. This representation supports the same conclusion: samples from BNPWMMD+DPMINE overlap more closely with real samples and better capture the underlying structure, indicating improved representation quality and reduced mode collapse.

To quantify dissimilarity between real and generated samples in the two-dimensional feature space, we consider the real features $\boldsymbol{f}_r := (\text{Feature1}_{1:200,r}, \text{Feature2}_{1:200,r})$ and the generated features $\boldsymbol{f}_g := (\text{Feature1}_{1:200,g}, \text{Feature2}_{1:200,g})$. We report the mean Fréchet Inception Distance (FID) and Kernel Inception Distance (KID) over 100 replications, together with variability measured as $\pm 2\sigma$, using both the t-SNE embedding and a custom encoder representation.

In addition, Table 2 reports empirical MMD values computed in the same two-dimensional feature space, using the same kernel and bandwidth parameters as those employed in the BNPWMMD training objective. We intentionally avoid reporting MMD in the original data space, since BNPWMMD-based models explicitly minimize data-space MMD during training, which would render such comparisons unfair.

Across all feature representations and metrics, BNPWMMD+DP–MINE consistently attains lower mean FID, KID, and MMD values than BNPWMMD alone. While the absolute differences are moderate and the $\pm 2\sigma$ intervals indicate non-negligible variability, the improvements are systematic across both t-SNE and custom-encoder embeddings. These trends are consistent with the visualizations in Figure 8 and suggest improved alignment between generated and real feature distributions, together with reduced mode collapse.

We further assess perceptual image quality using the Multi-Scale Structural Similarity Index (MS-SSIM) (Wang et al., 2003). MS-SSIM is reported as a mean $\pm 2\sigma$ over 100 replications, with higher values indicating better structural fidelity. Although the compared models are unconditional generators, the use of MS-SSIM is reasonable due to the strong anatomical alignment of axial, sagittal, and coronal lung CT slices across patients. The higher MS-SSIM score achieved by BNPWMMD+DP–MINE indicates improved visual consistency relative to BNPWMMD.

Additional implementation details are provided in Appendix E.3. Appendix E.2.2 further extends the evaluation to a brain MRI dataset, demonstrating that the observed improvements generalize across datasets and imaging modalities.

Table 2: Comparison of statistical scores (mean $\pm 2\sigma$) for the COVID-19 example over 100 replications.

| Evaluator | | BNPWMMD + DPMINE | BNPWMMD |
|---|---|---|---|
| FID | Custom Encoder | **0.00063 ± 0.00468** | 0.00084 ± 0.00535 |
| | t-SNE | **9.48447 ± 7.05185** | 11.10431 ± 10.8224 |
| KID | Custom Encoder | **0.00069 ± 0.00073** | 0.00361 ± 0.00102 |
| | t-SNE | **0.99858 ± 4.16996** | 1.011342 ± 6.66391 |
| MMD | Custom Encoder | **0.00902 ± 0.00236** | 0.01172 ± 0.00703 |
| | t-SNE | **0.00074 ± 0.00130** | 0.00358 ± 0.00782 |
| MS-SSIM | – | **0.43465 ± 0.02189** | 0.41781 ± 0.06755 |

## 5 Concluding Remarks

In this work, a Bayesian bootstrap resampling framework was introduced for mutual information estimation via MINE (DPMINE), enabling variational lower-bound optimization within Bayesian nonparametric

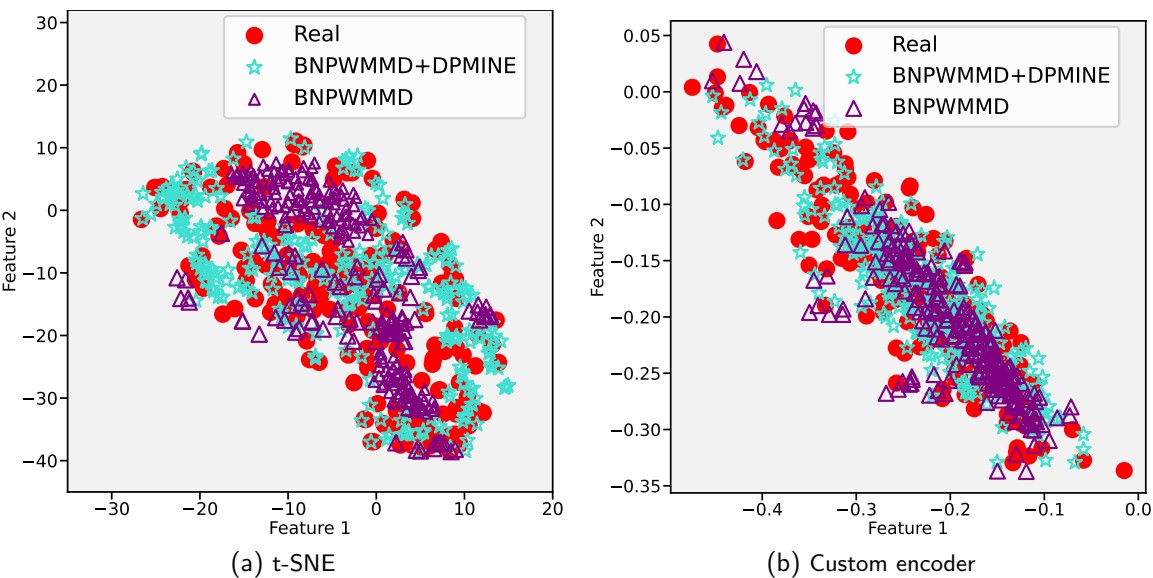

Figure 8: Scatter plots of two-dimensional feature representations in the COVID-19 example, comparing samples generated using BNPWMMD+DPMINE with those generated using BNPWMMD. Panel (a) shows the t-SNE embedding of the features, while panel (b) displays the features obtained from a custom encoder.

learning (BNPL). By replacing uniform weights with Dirichlet-resampled weights, the proposed approach naturally accounts for uncertainty in the underlying data distribution, a core feature of BNPL.

The resulting estimator, DPMINE, yields tighter and lower-variance MI estimates than standard MINE and performs competitively with existing variational bounds. Empirical results show stable behavior across batch sizes and high-dimensional settings. When incorporated into a BNP VAE-GAN model, DPMINE improves training stability and generative quality, demonstrating its practical value.

Overall, this work bridges an important gap between MI-based learning and BNPL and enables broader use of information-theoretic objectives in Bayesian nonparametric frameworks. While the focus here is on generative modeling, the proposed framework is applicable to other learning problems, including representation learning and Bayesian decision-making.

Extending DPMINE to non-IID settings and distributed or privacy-sensitive learning scenarios remains an important direction for future work. In particular, incorporating DPMINE into federated learning frameworks could strengthen privacy-preserving training by providing more robust and stable control of information flow across decentralized and heterogeneous data sources. However, the current formulation relies on an IID assumption underlying the Bayesian bootstrap. Adapting DPMINE to federated settings will therefore require new methodological developments to handle client-level heterogeneity, non-IID sampling, and communication constraints while preserving theoretical guarantees.

## Acknowledgments

Bei Jiang and Linglong Kong were partially supported by grants from the Canada CIFAR AI Chairs program, the Alberta Machine Intelligence Institute (AMII), and Natural Sciences and Engineering Council of Canada (NSERC), and Linglong Kong was also partially supported by grants from the Canada Research Chair program from NSERC. Michael Zhang was supported by the University of Hong Kong Seed Fund for PI Research #2402101367. The authors would also like to thank the anonymous reviewers for their constructive comments that improved the quality of this article.

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

## Appendix

## A  Notations

- $F$: Real data distribution.

- $(\mathbf{X}_{1:n})$: A sample of $n$ independent and identically distributed random variables generated from $F$.

- $\mathrm{DP}(a + n, H_n^*)$: The Dirichlet process (DP) posterior with concentration parameter $a + n$, $a > 0$, and base measure $H_n^*$.

- $H_n^*$: A mixture probability measure given by $\frac{a}{a+n} H + \frac{n}{a+n} F_{\mathbf{x}_{1:n}}$.

- $H$: The base measure of the DP prior $\mathrm{DP}(a, H)$.

- $F_{\mathbf{x}_{1:n}}$: The empirical probability measure based on the sample data defined by $\frac{1}{n} \sum_{i=1}^n \delta_{\mathbf{X}_i}$.

- $F_N^{\mathrm{Pos}}$: The probability measure of the DP posterior approximation defined by $\sum_{i=1}^N J_i^{\mathrm{Pos}} \delta_{\mathbf{X}_i^{\mathrm{Pos}}}$ with $(\mathbf{X}_{1:N}^{\mathrm{Pos}}) \sim H_n^*$ and $(J_{1:N}^{\mathrm{Pos}}) \sim \mathrm{Dirichlet}((a + n)/N, \ldots, (a + n)/N)$ Ishwaran & Zarepour (2002).

- $f_i(\mathbf{X})$: A continuous function of a random variable $\mathbf{X} \sim F$ for $i = 1, 2$.

- $\{T_{\boldsymbol{\gamma}}\}_{\boldsymbol{\gamma} \in \boldsymbol{\Gamma}}$: A set of continuous functions parameterized by a neural network on a compact domain $\boldsymbol{\Gamma}$.

- $I_d$: Identity matrix of size $d \times d$.

- "a.s.": Standing for "almost surely" and indicates that the statements hold with probability 1.

## B  Definition of MMD and Wasserstein Distances: DP and Empirical Representations

Selecting an appropriate statistical distance is crucial for effective generative model training. Here, we focus on the DP representation of two popular distances used in BNP deep learning and we will briefly mention their frequentist counterparts.

### B.1  Maximum Mean Discrepancy Distance (Feature-Matching Comparison)

The MMD distance was initially introduced in Gretton et al. (2012) for frequentist two-sample comparisons. Recently, a DP-based version of this distance has been proposed for BNP hypothesis testing Fazeli-Asl et al. (2024). Consider a set of functions $\{G_{\boldsymbol{\omega}}\}_{\boldsymbol{\omega} \in \boldsymbol{\Omega}}$ parameterized by a neural network that can generate $n$ IID random variables $(\mathbf{Y}_{1:n})$, where the likelihood function is intractable and not accessible. Let $k(\cdot, \cdot)$ be a continuous kernel function with a feature space corresponding to a universal reproducing kernel Hilbert space Gretton et al. (2012) defined on a compact sample space $\mathfrak{X}$ Gretton et al. (2012). Given a sample

$(\mathbf{X}_{1:n}) \overset{\text{iid}}{\sim} F$, the MMD distance between $F^{\text{Pos}}$ and $F_{\mathbf{Y}_{1:n}}$ is approximated as:

$$\text{MMD}^2(F_N^{\text{Pos}}, F_{\mathbf{Y}_{1:n}}) := \sum_{\ell,t=1}^{N} J_\ell^{\text{Pos}} J_t^{\text{Pos}} k(\mathbf{X}_\ell^{\text{Pos}}, \mathbf{X}_t^{\text{Pos}})$$

$$- \frac{2}{n} \sum_{\ell=1}^{N} \sum_{t=1}^{n} J_\ell^{\text{Pos}} k(\mathbf{X}_\ell^{\text{Pos}}, \mathbf{Y}_t) + \frac{1}{n^2} \sum_{\ell,t=1}^{n} k(\mathbf{Y}_\ell, \mathbf{Y}_t). \quad (26)$$

In the frequentist version of the MMD distance, as defined in Gretton et al. (2012), the empirical distribution $F_{\mathbf{X}_{1:n}}$ is considered. This is denoted as $\text{MMD}^2(F_{\mathbf{X}_{1:n}}, F_{\mathbf{Y}_{1:n}})$, which is obtained by replacing $N$, $J_{1:N}^{\text{Pos}}$, and $\mathbf{X}_{1:N}^{\text{Pos}}$ with $n$, $1/n$, and $\mathbf{X}_{1:n}$, respectively, in equation 26.

### B.2   Wasserstein Distance (Overall Distribution Comparison)

The frequentist version of the Wasserstein distance is completely discussed in (Villani, 2008, Part I6). Fazeli-Asl et al. Fazeli-Asl & Zhang (2023) proposed a BNP version of this distance through its Kantorovich-Rubinstein dual representation. Let $\{D_{\boldsymbol{\theta}}\}_{\boldsymbol{\theta} \in \Theta}$ be a parametrized family of continuous functions that all are 1-Lipschitz. Then the Wasserstein distance between $F^{\text{Pos}}$ and $F_{\mathbf{Y}_{1:n}}$ is approximated as:

$$\text{WS}(F_N^{\text{Pos}}, F_{\mathbf{Y}_{1:n}}) := \max_{\Theta} \sum_{i=1}^{N} \left( J_i^{\text{Pos}} D_\theta(\mathbf{X}_i^{\text{Pos}}) - \frac{D_{\boldsymbol{\theta}}(\mathbf{Y}_i)}{n} \right). \quad (27)$$

By utilizing modifications in equation 27 similar to those described for the MMD distance, the empirical representation of the Wasserstein distance can also be obtained. In this section, we propose two novel representations for the MINE using the DP. These representations are based on the KL and JS divergences will be used in our BNP learning framework to maximize information during the training process.

## C   BNPWMMD-GAN

The BNPWMMD-GAN architecture introduced by Fazeli-Asl & Zhang (2023) consists of four networks: an encoder $\{E_{\boldsymbol{\eta}}\}_{\boldsymbol{\eta} \in \mathcal{H}}$, a generator $\{G_{\boldsymbol{\omega}}\}_{\boldsymbol{\omega} \in \Omega}$, a code generator $\{CG_{\boldsymbol{\omega}'}\}_{\boldsymbol{\omega}' \in \Omega'}$, and a discriminator $\{D_{\boldsymbol{\theta}}\}_{\boldsymbol{\theta} \in \Theta}$.

Given the codes $\left(\boldsymbol{c}_{1:N} := E_{\boldsymbol{\eta}}\left(\mathbf{X}_{1:N}^{\text{Pos}}\right)\right)$ and generated codes $\left(\widetilde{\boldsymbol{c}}_{1:N} := CG_{\boldsymbol{\omega}'}\left(\boldsymbol{\varepsilon}'_{1:N}\right)\right)$ along with the latent noise $(\boldsymbol{\varepsilon}_{1:N} \in \mathbb{R}^p) \sim F_{\boldsymbol{\varepsilon}}$ and sub-latent noise $(\boldsymbol{\varepsilon}'_{1:N} \in \mathbb{R}^q) \sim F_{\boldsymbol{\varepsilon}'}$, where $q < p$, the BNPWMMD-GAN is trained by updating the networks' parameters according to the following hybrid objective function[2]:

$$(\hat{\boldsymbol{\omega}}, \hat{\boldsymbol{\eta}})(F^{\text{Pos}}) = \underset{\boldsymbol{\Omega}, \mathcal{H}}{\arg\min} \left\{ -\frac{1}{N} \sum_{i=1}^{N} \Big[ D_{\boldsymbol{\theta}}(G_{\boldsymbol{\omega}}(\varepsilon_i)) + D_{\boldsymbol{\theta}}(G_{\boldsymbol{\omega}}(c_i)) + D_{\boldsymbol{\theta}}(G_{\boldsymbol{\omega}}(\tilde{c}_i)) \Big] + \text{MMD}(F_N^{\text{Pos}}, F_{G_{\boldsymbol{\omega}}(\boldsymbol{\varepsilon}_{1:N})}) \right.$$

$$\left. \underbrace{-\text{MMD}(F_N^{\text{Pos}}, F_{G_{\boldsymbol{\omega}}(\tilde{c}_{1:N})}) + \text{MMD}(F_N^{\text{Pos}}, F_{G_{\boldsymbol{\omega}}(c_{1:N})})}_{\mathcal{I}_1(\boldsymbol{\omega}, \boldsymbol{\eta})} + \underbrace{\text{MMD}(F_{\boldsymbol{\varepsilon}_{1:N}}, F_{c_{1:N}})}_{\mathcal{I}_2(\boldsymbol{\omega}, \boldsymbol{\eta})} \right\}, \quad (28a)$$

$$\hat{\boldsymbol{\theta}}(F^{\text{Pos}}) = \underset{\boldsymbol{\Theta}}{\arg\min} \left\{ \underbrace{\frac{1}{N} \sum_{i=1}^{N} \Big[ D_{\boldsymbol{\theta}}(G_{\boldsymbol{\omega}}(\varepsilon_i)) + D_{\boldsymbol{\theta}}(G_{\boldsymbol{\omega}}(c_i)) + D_{\boldsymbol{\theta}}(G_{\boldsymbol{\omega}}(\tilde{c}_i)) - 3J_i^{\text{Pos}} D_{\boldsymbol{\theta}}(\mathbf{X}_i^{\text{Pos}}) \Big]}_{\mathcal{J}_1(\boldsymbol{\theta})} + \lambda \underbrace{L_{\text{GP-D}}}_{\mathcal{J}_2(\boldsymbol{\theta})} \right\},$$

$$(28b)$$

$$\hat{\boldsymbol{\omega}}'(F^{\text{Pos}}) = \underset{\boldsymbol{\Omega}'}{\arg\min} \, \text{MMD}(F_{c_{1:N}}, F_{\tilde{c}_{1:N}}). \quad (28c)$$

---

[2]Each $F$ indexed by a sample vector in equation 28 indicates the corresponding empirical distribution.

Simultaneous updates of parameters $\boldsymbol{\omega}$ and $\boldsymbol{\eta}$ are facilitated by the essential role played by the generator function, which serves as a decoder during VAE training. Here, the generator is fed with $(\boldsymbol{\varepsilon}_{1:N}, \boldsymbol{c}_{1:N}, \widetilde{\boldsymbol{c}}_{1:N})$, and terms $\mathcal{I}_1(\boldsymbol{\omega}, \boldsymbol{\eta})$ and $\mathcal{J}_1(\boldsymbol{\theta})$ refer to minimizing the combined distance given by:

$$d_{\mathrm{WMMD}}(F^{\mathrm{Pos}}, F_{G_{\boldsymbol{\omega}}}) = \mathrm{WS}(F^{\mathrm{Pos}}, F_{G_{\boldsymbol{\omega}}}) + \mathrm{MMD}(F^{\mathrm{Pos}}, F_{G_{\boldsymbol{\omega}}}). \tag{29}$$

Since the MMD measure is an $L^2$-norm distance, the MMD-based terms in $\mathcal{I}_1(\boldsymbol{\omega}, \boldsymbol{\eta})$ serves as the posterior reconstruction errors, while $\mathcal{I}_2(\boldsymbol{\omega}, \boldsymbol{\eta})$ serves as the regularization error for approximating the variational distribution $F_{E_{\boldsymbol{\eta}}}(\boldsymbol{c}_{1:N} | \mathbf{X}_{1:N}^{\mathrm{Pos}})$. Furthermore, the gradient penalty $\mathcal{J}_2(\boldsymbol{\theta})$ with a positive coefficient $\lambda$ was used to ensure training stability by forcing the 1-Lipschitz constraint on the discriminator (Gulrajani et al., 2017). Specifically, this penalty is given by

$$L_{\mathrm{GP-D}} = \frac{1}{N} \sum_{i=1}^{N} (\|\nabla_{\widehat{\mathbf{X}}_i^{\mathrm{Pos}}} D_{\boldsymbol{\theta}}(\widehat{\mathbf{X}}_i^{\mathrm{Pos}})\|_2 - 1)^2, \tag{30}$$

where $\widehat{\mathbf{X}}_i^{\mathrm{Pos}} = u\widetilde{\mathbf{X}}_i^{\mathrm{Pos}} + (1-u)\mathbf{X}_i^{\mathrm{Pos}}$ and $\widetilde{\mathbf{X}}_i^{\mathrm{Pos}}$ represents any posterior fake sample generated by the generator.

## D  Theoretical Proofs

**Theorem 1.**   Consider the DP posterior representations of MINE presented in the main paper. Assume that both $T_{\boldsymbol{\gamma}}$ and $e^{T_{\boldsymbol{\gamma}}}$ are $M$-bounded for all $\boldsymbol{\gamma} \in \boldsymbol{\Gamma}$ (i.e., $|T_{\boldsymbol{\gamma}}|, |e^{T_{\boldsymbol{\gamma}}}| \leq M$). Given the DP posterior approximation in equation 7, for any $N > 0$, we have,

    *i.* $\displaystyle \lim_{n \to \infty} \mathbb{E}_{F_N^{\mathrm{Pos}}} \left( L_{\widehat{\boldsymbol{\gamma}}}^{\mathrm{DPKL}}(f_1(\mathbf{X}_{1:N}^{\mathrm{Pos}}), f_2(\mathbf{X}_{1:N}^{\mathrm{Pos}})) \right) \geq L_{\widehat{\boldsymbol{\gamma}}}^{\mathrm{KL}}(\mathbf{X}_1', \mathbf{X}_2'),$

    *ii.* $\displaystyle \lim_{n \to \infty} \mathbb{E}_{F_N^{\mathrm{Pos}}} \left( L_{\widehat{\boldsymbol{\gamma}}}^{\mathrm{DPJS}}(f_1(\mathbf{X}_{1:N}^{\mathrm{Pos}}), f_2(\mathbf{X}_{1:N}^{\mathrm{Pos}})) \right) \geq L_{\boldsymbol{\gamma}}^{\mathrm{JS}}(\mathbf{X}_1', \mathbf{X}_2').$

*Proof.* Recall that, for all $\boldsymbol{\gamma} \in \boldsymbol{\Gamma}$,

$$L_{\boldsymbol{\gamma}}^{\mathrm{DPKL}}(f_1(\mathbf{X}_{1:N}^{\mathrm{Pos}}), f_2(\mathbf{X}_{1:N}^{\mathrm{Pos}})) := \sum_{\ell=1}^{N} J_{\ell}^{\mathrm{Pos}} T_{\boldsymbol{\gamma}}(f_1(\mathbf{X}_{\ell}^{\mathrm{Pos}}), f_2(\mathbf{X}_{\ell}^{\mathrm{Pos}})) - \ln \sum_{\ell=1}^{N} J_{\ell}^{\mathrm{Pos}} e^{T_{\boldsymbol{\gamma}}(f_1(\mathbf{X}_{\ell}^{\mathrm{Pos}}), f_2(\mathbf{X}_{\pi(\ell)}^{\mathrm{Pos}}))}, \tag{31}$$

where $\left(J_{1:N}^{\mathrm{Pos}}\right) \sim \mathrm{Dirichlet}((a+n)/N, \ldots, (a+n)/N)$ and $\left(\mathbf{X}_{1:N}^{\mathrm{Pos}}\right) \overset{\mathrm{iid}}{\sim} H_n^*$.

Applying the law of total expectation yields:

$$\mathbb{E}_{F^{\mathrm{Pos}}} \left( L_{\boldsymbol{\gamma}}^{\mathrm{DPKL}} \left( f_1(\mathbf{X}_{1:N}^{\mathrm{Pos}}), f_2(\mathbf{X}_{1:N}^{\mathrm{Pos}}) \right) \right) = \mathbb{E}_{H_n^*} \left( \mathbb{E}_{F_{\boldsymbol{J}^{\mathrm{Pos}}}} \left( L_{\boldsymbol{\gamma}}^{\mathrm{DPKL}} \left( f_1(\mathbf{X}_{1:N}^{\mathrm{Pos}}), f_2(\mathbf{X}_{1:N}^{\mathrm{Pos}}) \right) \mid \mathbf{X}_{1:N}^{\mathrm{Pos}} \right) \right), \tag{32}$$

where $F_{\boldsymbol{J}^{\mathrm{Pos}}}$ is the CDF corresponding to $\boldsymbol{J}^{\mathrm{Pos}} := \left(J_{1:N}^{\mathrm{Pos}}\right)$.

Taking the expectation in equation 32 with respect to the Dirichlet weights gives:

$$\mathbb{E}_{F_{\boldsymbol{J}^{\mathrm{Pos}}}}\left(L_{\boldsymbol{\gamma}}^{\mathrm{DPKL}}\left(f_1(\mathbf{X}_{1:N}^{\mathrm{Pos}}), f_2(\mathbf{X}_{1:N}^{\mathrm{Pos}})\right) \mid \mathbf{X}_{1:N}^{\mathrm{Pos}}\right) = \sum_{\ell=1}^{N} \mathbb{E}_{F_{\boldsymbol{J}^{\mathrm{Pos}}}}\left(J_{\ell}^{\mathrm{Pos}}\right) T_{\boldsymbol{\gamma}}(f_1(\mathbf{X}_{\ell}^{\mathrm{Pos}}), f_2(\mathbf{X}_{\ell}^{\mathrm{Pos}}))$$

$$- \mathbb{E}_{F_{\boldsymbol{J}^{\mathrm{Pos}}}}\left(\ln \sum_{\ell=1}^{N} J_{\ell}^{\mathrm{Pos}} e^{T_{\boldsymbol{\gamma}}(f_1(\mathbf{X}_{\ell}^{\mathrm{Pos}}), f_2(\mathbf{X}_{\pi(\ell)}^{\mathrm{Pos}}))}\right) \qquad (33\mathrm{a})$$

(Linearity of Expectation)

$$\geq \sum_{\ell=1}^{N} \mathbb{E}_{F_{\boldsymbol{J}^{\mathrm{Pos}}}}\left(J_{\ell}^{\mathrm{Pos}}\right) T_{\boldsymbol{\gamma}}(f_1(\mathbf{X}_{\ell}^{\mathrm{Pos}}), f_2(\mathbf{X}_{\ell}^{\mathrm{Pos}}))$$

$$- \ln \sum_{\ell=1}^{N} \mathbb{E}_{F_{\boldsymbol{J}^{\mathrm{Pos}}}}\left(J_{\ell}^{\mathrm{Pos}}\right) e^{T_{\boldsymbol{\gamma}}(f_1(\mathbf{X}_{\ell}^{\mathrm{Pos}}), f_2(\mathbf{X}_{\pi(\ell)}^{\mathrm{Pos}}))} \qquad (33\mathrm{b})$$

(Jensen's Inequality)

$$\geq \sum_{\ell=1}^{N} \frac{1}{N} T_{\boldsymbol{\gamma}}(f_1(\mathbf{X}_{\ell}^{\mathrm{Pos}}), f_2(\mathbf{X}_{\ell}^{\mathrm{Pos}}))$$

$$- \ln \sum_{\ell=1}^{N} \frac{1}{N} e^{T_{\boldsymbol{\gamma}}(f_1(\mathbf{X}_{\ell}^{\mathrm{Pos}}), f_2(\mathbf{X}_{\pi(\ell)}^{\mathrm{Pos}}))} \qquad (33\mathrm{c})$$

(Dirichlet Weights).

Taking the expectation in equation 33 with respect to the posterior atoms gives

$$\mathbb{E}_{H_n^*}\left(\mathbb{E}_{F_{\boldsymbol{J}^{\mathrm{Pos}}}}\left(L_{\boldsymbol{\gamma}}^{\mathrm{DPKL}}\left(f_1(\mathbf{X}_{1:N}^{\mathrm{Pos}}), f_2(\mathbf{X}_{1:N}^{\mathrm{Pos}})\right) \mid \mathbf{X}_{1:N}^{\mathrm{Pos}}\right)\right) \geq \sum_{\ell=1}^{N} \frac{1}{N} \mathbb{E}_{H_n^*}\left(T_{\boldsymbol{\gamma}}(f_1(\mathbf{X}_{\ell}^{\mathrm{Pos}}), f_2(\mathbf{X}_{\ell}^{\mathrm{Pos}}))\right)$$

$$- \mathbb{E}_{H_n^*}\left(\ln \sum_{\ell=1}^{N} \frac{1}{N} e^{T_{\boldsymbol{\gamma}}(f_1(\mathbf{X}_{\ell}^{\mathrm{Pos}}), f_2(\mathbf{X}_{\pi(\ell)}^{\mathrm{Pos}}))}\right) \qquad (34\mathrm{a})$$

(Linearity of Expectation)

$$\geq \sum_{\ell=1}^{N} \frac{1}{N} \mathbb{E}_{H_n^*}\left(T_{\boldsymbol{\gamma}}(f_1(\mathbf{X}_{\ell}^{\mathrm{Pos}}), f_2(\mathbf{X}_{\ell}^{\mathrm{Pos}}))\right)$$

$$- \ln \sum_{\ell=1}^{N} \frac{1}{N} \mathbb{E}_{H_n^*}\left(e^{T_{\boldsymbol{\gamma}}(f_1(\mathbf{X}_{\ell}^{\mathrm{Pos}}), f_2(\mathbf{X}_{\pi(\ell)}^{\mathrm{Pos}}))}\right) \qquad (34\mathrm{b})$$

(Jensen's Inequality)

$$= \mathbb{E}_{H_n^*}\left(T_{\boldsymbol{\gamma}}(f_1(\mathbf{X}_{\ell}^{\mathrm{Pos}}), f_2(\mathbf{X}_{\ell}^{\mathrm{Pos}}))\right)$$

$$- \ln \mathbb{E}_{H_n^*}\left(e^{T_{\boldsymbol{\gamma}}(f_1(\mathbf{X}_{\ell}^{\mathrm{Pos}}), f_2(\mathbf{X}_{\pi(\ell)}^{\mathrm{Pos}}))}\right) \qquad (34\mathrm{c})$$

(Identical Atoms).

On the other hand, as $n$ approaches infinity, the Glivenko–Cantelli theorem implies that $F_{\mathbf{X}_{1:n}}$ converges to $F$ and subsequently $H_n^*$ converges to $F$. This convergence implies that

$$\mathbf{X}_{\ell}^{\mathrm{pos}} \xrightarrow{d} \mathbf{X}_{\ell}, \qquad (35)$$

where "$\xrightarrow{d}$" denotes convergence in distribution and $\mathbf{X}_{\ell}$ is a random variable distributed according to $F$, for $\ell = 1, \ldots, N$. By applying the continuous mapping theorem, we have

$$T_{\boldsymbol{\gamma}}(f_1(\mathbf{X}_{\ell}^{\mathrm{Pos}}), f_2(\mathbf{X}_{\ell}^{\mathrm{Pos}})) \xrightarrow{d} T_{\boldsymbol{\gamma}}(f_1(\mathbf{X}_{\ell}), f_2(\mathbf{X}_{\ell})), \qquad e^{T_{\boldsymbol{\gamma}}(f_1(\mathbf{X}_{\ell}^{\mathrm{Pos}}), f_2(\mathbf{X}_{\pi(\ell)}^{\mathrm{Pos}}))} \xrightarrow{d} e^{T_{\boldsymbol{\gamma}}(f_1(\mathbf{X}_{\ell}), f_2(\mathbf{X}_{\pi(\ell)}))}. \qquad (36)$$

Since $T_{\boldsymbol{\gamma}}$ and $e^{T_{\boldsymbol{\gamma}}}$ are $M$-bounded, it implies

$$\mathbb{E}_{H_n^*}\left(T_{\boldsymbol{\gamma}}(f_1(\mathbf{X}_\ell^{\text{Pos}}), f_2(\mathbf{X}_\ell^{\text{Pos}}))\right) \to \mathbb{E}_F\left(T_{\boldsymbol{\gamma}}(f_1(\mathbf{X}_\ell), f_2(\mathbf{X}_\ell))\right), \tag{37}$$

$$\mathbb{E}_{H_n^*}\left(e^{T_{\boldsymbol{\gamma}}(f_1(\mathbf{X}_\ell^{\text{Pos}}), f_2(\mathbf{X}_{\pi(\ell)}^{\text{Pos}}))}\right) \to \mathbb{E}_F\left(e^{T_{\boldsymbol{\gamma}}(f_1(\mathbf{X}_\ell), f_2(\mathbf{X}_{\pi(\ell)}))}\right). \tag{38}$$

Finally, taking the limit in equation 34 as $n \to \infty$ and applying the continuous mapping theorem implies

$$\lim_{n\to\infty} \mathbb{E}_{H_n^*}\left(\mathbb{E}_{F_{\boldsymbol{J}^{\text{Pos}}}}\left(L_{\boldsymbol{\gamma}}^{\text{DPKL}}\left(f_1(\mathbf{X}_{1:N}^{\text{Pos}}), f_2(\mathbf{X}_{1:N}^{\text{Pos}})\right) \mid \mathbf{X}_{1:N}^{\text{Pos}}\right)\right) \geq \mathbb{E}_F\left(T_{\boldsymbol{\gamma}}(f_1(\mathbf{X}_\ell), f_2(\mathbf{X}_\ell))\right)$$

$$- \ln \mathbb{E}_F\left(e^{T_{\boldsymbol{\gamma}}(f_1(\mathbf{X}_\ell), f_2(\mathbf{X}_{\pi(\ell)}))}\right) \tag{39}$$

$$= \mathbb{E}_{F_{\mathbf{X}_1' \mathbf{X}_2'}}\left(T_{\boldsymbol{\gamma}}(\mathbf{X}_1', \mathbf{X}_2')\right)$$

$$- \ln \mathbb{E}_{F_{\mathbf{X}_1'} \otimes F_{\mathbf{X}_2'}}\left(e^{T_{\boldsymbol{\gamma}}(\mathbf{X}_1', \mathbf{X}_2')}\right) \tag{40}$$

$$= L_{\boldsymbol{\gamma}}^{\text{KL}}(\mathbf{X}_1', \mathbf{X}_2') \tag{41}$$

with $\mathbf{X}_1' = f_1(\mathbf{X}_\ell)$ and $\mathbf{X}_2' = f_2(\mathbf{X}_\ell)$. Consequently, for all $\boldsymbol{\gamma} \in \boldsymbol{\Gamma}$,

$$\lim_{n\to\infty} \mathbb{E}_{F_N^{\text{Pos}}}\left[L_{\boldsymbol{\gamma}}^{\text{DPKL}}(f_1(\mathbf{X}_{1:N}^{\text{Pos}}), f_2(\mathbf{X}_{1:N}^{\text{Pos}}))\right] \geq L_{\boldsymbol{\gamma}}^{\text{KL}}(\mathbf{X}_1', \mathbf{X}_2'). \tag{42}$$

Moreover, since

$$\max_{\boldsymbol{\gamma} \in \boldsymbol{\Gamma}} L_{\boldsymbol{\gamma}}^{\text{DPKL}}(f_1(\mathbf{X}_{1:N}^{\text{Pos}}), f_2(\mathbf{X}_{1:N}^{\text{Pos}})) \geq L_{\boldsymbol{\gamma}}^{\text{DPKL}}(f_1(\mathbf{X}_{1:N}^{\text{Pos}}), f_2(\mathbf{X}_{1:N}^{\text{Pos}})) \tag{43}$$

for any $\boldsymbol{\gamma} \in \boldsymbol{\Gamma}$, taking expectations and limits yields

$$\lim_{n\to\infty} \mathbb{E}_{F_N^{\text{Pos}}}\left[\max_{\boldsymbol{\gamma} \in \boldsymbol{\Gamma}} L_{\boldsymbol{\gamma}}^{\text{DPKL}}(f_1(\mathbf{X}_{1:N}^{\text{Pos}}), f_2(\mathbf{X}_{1:N}^{\text{Pos}}))\right] \geq L_{\boldsymbol{\gamma}}^{\text{KL}}(\mathbf{X}_1', \mathbf{X}_2'). \tag{44}$$

Since the above inequality holds for all $\boldsymbol{\gamma} \in \boldsymbol{\Gamma}$, it follows that

$$\lim_{n\to\infty} \mathbb{E}_{F_N^{\text{Pos}}}\left[\max_{\boldsymbol{\gamma} \in \boldsymbol{\Gamma}} L_{\boldsymbol{\gamma}}^{\text{DPKL}}(f_1(\mathbf{X}_{1:N}^{\text{Pos}}), f_2(\mathbf{X}_{1:N}^{\text{Pos}}))\right] \geq \max_{\boldsymbol{\gamma} \in \boldsymbol{\Gamma}} L_{\boldsymbol{\gamma}}^{\text{KL}}(\mathbf{X}_1', \mathbf{X}_2'), \tag{45}$$

which completes the proof of part (i).

A similar method is used to prove (ii) and it is then omitted.

$\square$

# E    Additional Results and Implementing Details

## E.1    DPMINE Evaluation on Complex Examples

In this section, we evaluate DPMINE against several benchmark methods on a set of complex examples adapted from Letizia et al. (2024), which are commonly used for assessing mutual information estimators.

### E.1.1    Complex Gaussian

**Gaussian Example.** Following the same protocol used in the main experimental section, a Gaussian dimension, staircase experiment is considered to evaluate mutual information estimators as the dimensionality of dependence increases. Let

$$\mathbf{X} = (\mathbf{U}, \mathbf{V}) \in \mathbb{R}^{2k}, \qquad \mathbf{U}, \mathbf{V} \in \mathbb{R}^k,$$

with $k \in \{1, 2, 3, 4, 5\}$, where each value of $k$ defines one staircase level.

For a fixed correlation parameter $\rho = 0.95$, the variables are generated according to the linear Gaussian relationship

$$\mathbf{V} = \rho\,\mathbf{U} + \sqrt{1-\rho^2}\,\boldsymbol{\varepsilon}, \qquad \boldsymbol{\varepsilon} \sim N(\mathbf{0}, \mathbf{I}_k),$$

with $\mathbf{U} \sim N(\mathbf{0}, \mathbf{I}_k)$ independent of $\boldsymbol{\varepsilon}$.

The true MI at each staircase level is given by

$$\mathrm{MI}(\mathbf{U}, \mathbf{V}) = -\frac{k}{2}\log\big(1-\rho^2\big),$$

and is used as a reference for comparing estimator performance in this example.

The cubic, half-cube, and asinh transformations introduced below induce heavy-tailed marginals while preserving the MI of this Gaussian example, since each is an invertible transformation applied componentwise.

**Cubic Example.** Let

$$\mathbf{U} \sim N(\mathbf{0}, \mathbf{I}_k),$$

and $\mathbf{V}$ is generated through a nonlinear transformation of a correlated Gaussian,

$$\mathbf{V} = \big(\rho\,\mathbf{U} + \sqrt{1-\rho^2}\,\boldsymbol{\varepsilon}\big)^3, \qquad \boldsymbol{\varepsilon} \sim N(\mathbf{0}, \mathbf{I}_k),$$

where the correlation parameter is fixed to $\rho = 0.95$.

**Half-Cube Example.** Let $\mathbf{U} \sim N(\mathbf{0}, \mathbf{I}_k)$. The variable $\mathbf{V}$ is first generated through a linear Gaussian relationship

$$\mathbf{Z} = \rho\,\mathbf{U} + \sqrt{1-\rho^2}\,\boldsymbol{\varepsilon}, \qquad \boldsymbol{\varepsilon} \sim N(\mathbf{0}, \mathbf{I}_k),$$

with $\rho = 0.95$, and is then transformed elementwise using a half-cube (tail-stretching) nonlinearity,

$$\mathbf{V} = \mathrm{sign}(\mathbf{Z})\,|\mathbf{Z}|^{3/2}.$$

**Asinh Example.** Let $\mathbf{U} \in \mathbb{R}^k$ be sampled from a standard multivariate Gaussian distribution, $\mathbf{U} \sim N(\mathbf{0}, \mathbf{I}_k)$. The variable $\mathbf{V}$ is generated by first forming a Gaussian vector

$$\mathbf{Z} = \rho\,\mathbf{U} + \sqrt{1-\rho^2}\,\boldsymbol{\varepsilon}, \qquad \boldsymbol{\varepsilon} \sim N(\mathbf{0}, \mathbf{I}_k),$$

and then applying the elementwise inverse hyperbolic sine (asinh) transformation,

$$\mathbf{V} = \mathrm{asinh}(\mathbf{Z}),$$

where $\mathrm{asinh}(\cdot)$ denotes the inverse hyperbolic sine function and $\rho = 0.95$ is fixed across all staircase levels.

### E.1.2 Non-Gaussian Distributions

**Student Example.** To examine estimator behavior under heavy-tailed dependence, a Student staircase example is considered. For each value of $k$, the joint vector $(\mathbf{U}, \mathbf{V})$ is sampled from a centered multivariate Student-$t$ distribution with the correlation structure

$$(\mathbf{U}, \mathbf{V}) \sim t_\nu\left(\mathbf{0}, \begin{pmatrix} \mathbf{I}_k & \rho\,\mathbf{I}_k \\ \rho\,\mathbf{I}_k & \mathbf{I}_k \end{pmatrix}\right),$$

where the degrees of freedom are fixed to $\nu = 2$ and the correlation parameter is set to $\rho = 0.95$.

The true MI at each staircase level is approximated via Monte Carlo integration of the corresponding Student-$t$ densities and serves as a reference throughout training.

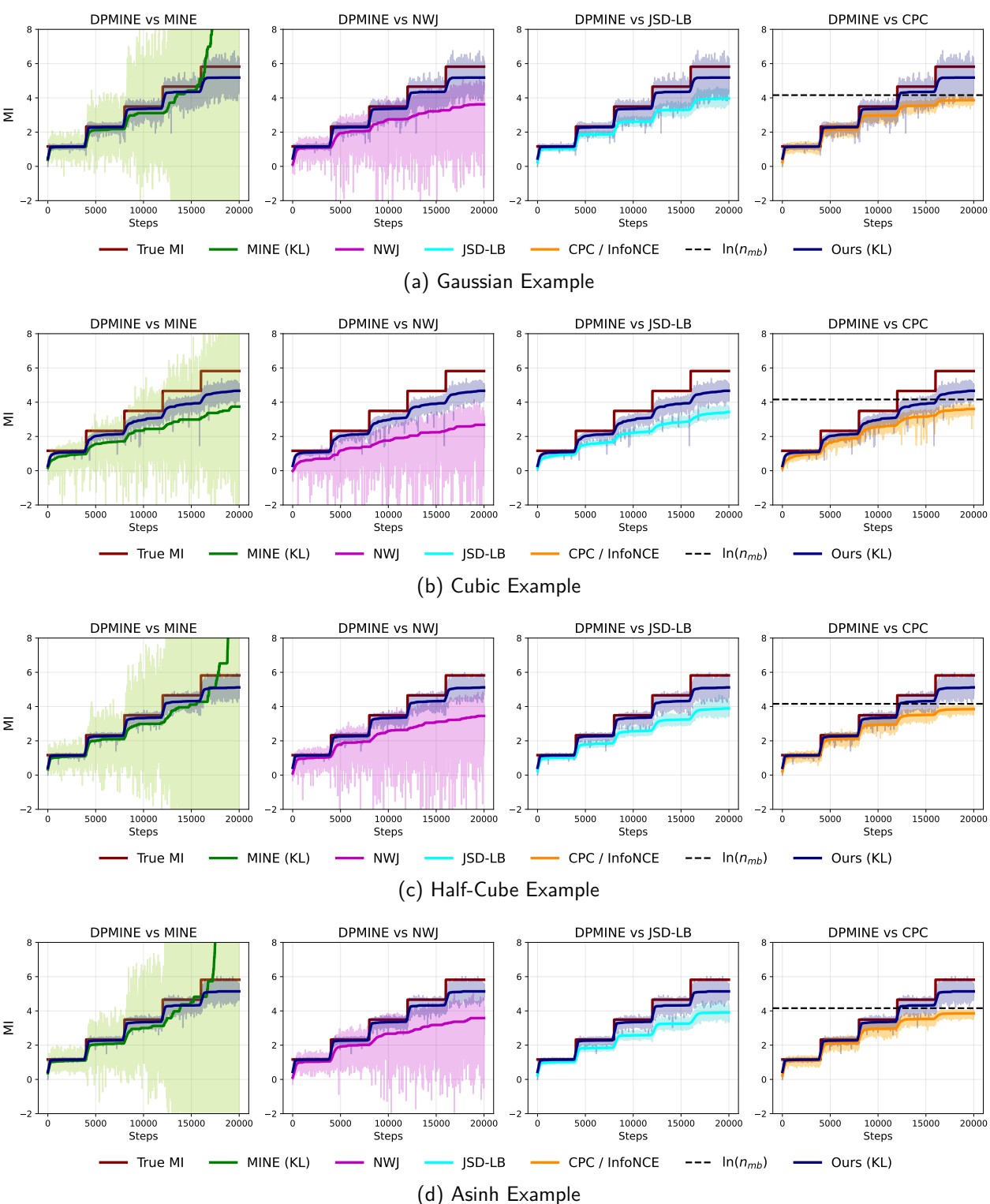

Figure 9: Complex Gaussian setting with $\mathbf{X} = (\mathbf{U}, \mathbf{V}) \sim F$. BNP-MINE uses mini-batch size 64, with $F \sim \mathrm{DP}(a_{\mathrm{MAP}}, H)$ and $H = N(\overline{\mathbf{X}}_{\mathrm{mb}}, S_{\mathbf{X}_{\mathrm{mb}}})$. Experiments are run for $k = 1, \ldots, 5$, using 4000 steps per dimension.

**Uniform Example.** A uniform-additive staircase example is used to study estimator behavior under bounded, non-Gaussian noise. At staircase level $k$, we consider a pair $(\mathbf{U}, \mathbf{V})$ with variable $\mathbf{U}$ is sampled uniformly on $[0, 1]^k$, and $\mathbf{V}$ is generated by adding independent uniform noise,

$$\mathbf{V} = \mathbf{U} + \boldsymbol{\varepsilon}, \qquad \boldsymbol{\varepsilon} \sim \mathrm{Unif}(-\varepsilon_0, \varepsilon_0)^k,$$

with $\varepsilon_0 = 0.05$ fixed across all staircase levels. Since the dependence is additive and independent across dimensions, the MI scales linearly with the dimension. The one-dimensional MI is approximated via numerical integration and scaled by $k$ to obtain the true MI at each staircase level.

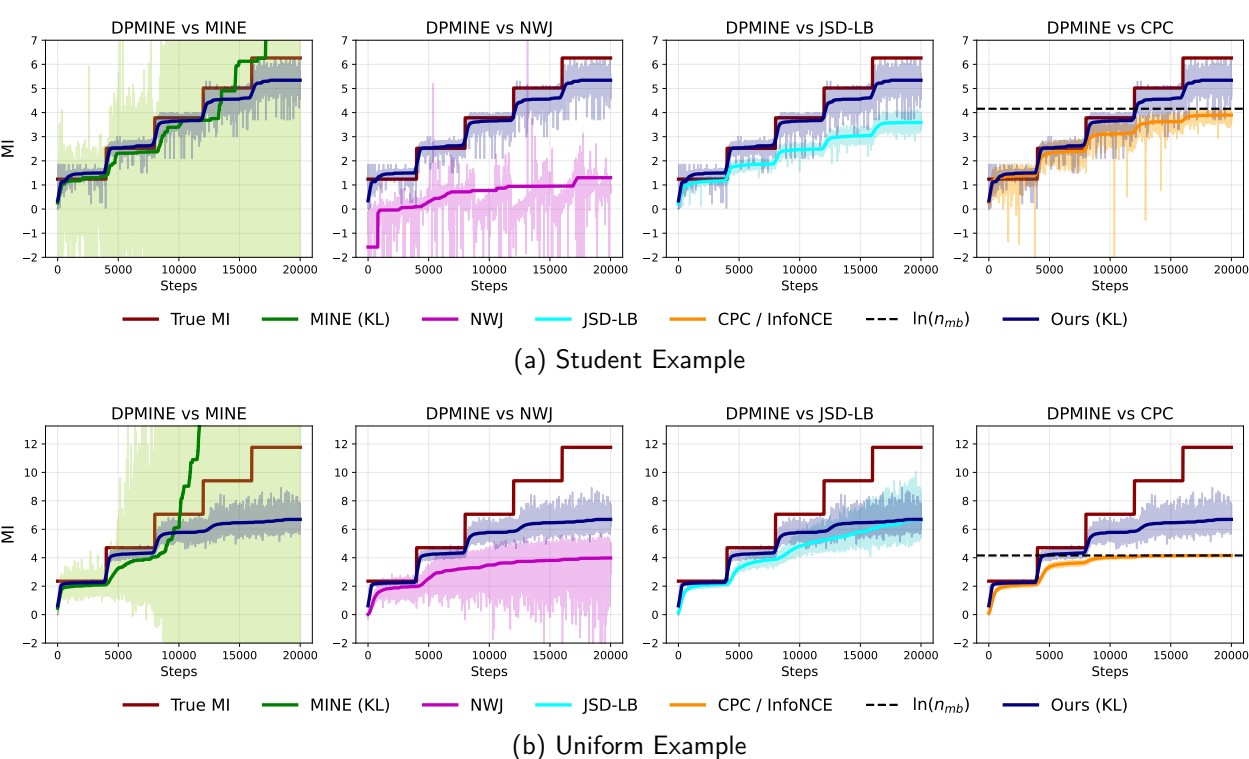

(a) Student Example

(b) Uniform Example

Figure 10: Complex non-Gaussian setting with $\mathbf{X} = (\mathbf{U}, \mathbf{V}) \sim F$. BNP-MINE uses mini-batch size 64, with $F \sim \mathrm{DP}(a_{\mathrm{MAP}}, H)$ and $H = N(\overline{\mathbf{X}}_{\mathrm{mb}}, S_{\mathbf{X}_{\mathrm{mb}}})$. Experiments are run for $k = 1, \ldots, 5$, using 4000 steps per dimension.

Figures 9 and 10 compare DPMINE with several variational MI estimators across Gaussian and non-Gaussian settings. Across all examples, DPMINE tracks the true MI more closely while exhibiting substantially reduced variance compared to MINE, NWJ, JSD-LB, and CPC/InfoNCE. Competing estimators either suffer from high variability or exhibit systematic underestimation and saturation effects, particularly as MI increases. In contrast, DPMINE shows stable and monotonic convergence toward the true MI across iterations, demonstrating improved robustness and reliability in both low- and high-MI regimes.

### E.2 Application of DPMINE (KL) in VAE-GAN

#### E.2.1 Toy Example

**Stanford Bunny Dataset:** We use the Stanford Bunny dataset, available at `https://graphics.stanford.edu/data/3Dscanrep/`, to investigate the impact of DPMINE on an additional toy example. The dataset consists of point clouds representing the Stanford Bunny, a renowned 3D model provided by the Stanford University Computer Graphics Laboratory. These point clouds, captured with the Cyberware 3030 MS scanner and stored in PLY files (Polygon File Format) developed at Stanford, represent spatial locations on the object's surface. Point clouds provide a 3D spatial representation of the object, enabling

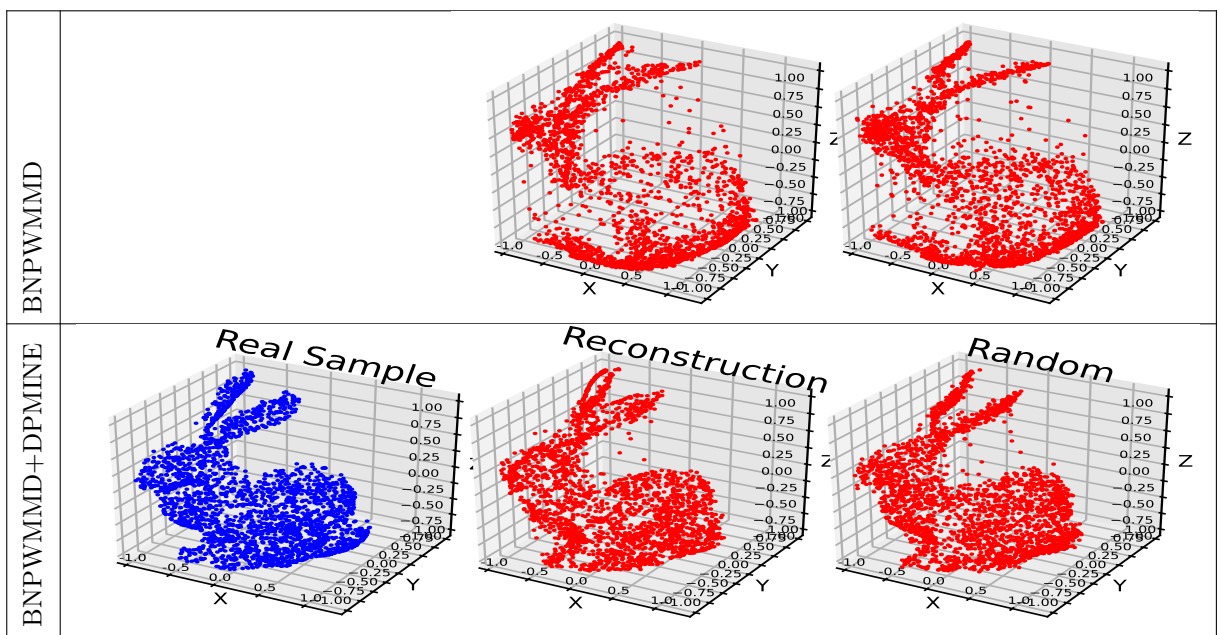

Figure 11: 2500 samples of the Stanford Bunny generated using BNPWMMD+DPMINE after 5000 epochs, compared with BNPWMMD alone.

detailed visualization and analysis. For our study, we use three point clouds from this dataset and apply filtering with the *pyoints* Python library to extract points in 3D space.

We randomly sampled 5000 points from the available 43,188 points as the training dataset and implemented our model on them by feeding the model with noise inputs to generate 2500 random samples and with encoded real inputs to obtain reconstruction samples. Figure 11 illustrates the significant impact of DPMINE on the performance of the BNP VAE-GAN in data generation.

### E.2.2   Real Examples

**Covid-19 dataset:**   The red border in Figure 12 indicates the corresponding slices depicted in Figure 7 of the experimental findings discussed in the main paper. The results indicate that BNPWMMD+DPMINE yields improved sharpness and diversity in the generated 3D slices relative to the baseline methods. Figure 13 further shows reconstructed samples that exhibit increased similarity to the training data under the BNPWMMD+DPMINE model.

**BraTS 2018 Dataset:**   The BRATS 2018 dataset, a benchmark resource in medical imaging available at `https://www.med.upenn.edu/sbia/brats2018/data.html`, is employed for training models in the generation of brain tumor MRI scans. For the experiments, data from 210 subjects labeled as "HGG" (High-Grade Glioma) are used, focusing on patients with aggressive brain tumors. Each subject's MRI data includes four distinct imaging modalities: T1-weighted (T1), T1-weighted with contrast enhancement (T1ce), T2-weighted (T2), and Fluid Attenuated Inversion Recovery (FLAIR). Notably, the FLAIR modality, which is particularly effective for highlighting edema and tumor boundaries, is used for the experiments, providing essential insights for synthetic MRI generation.

The results presented in Figures 14-17 and Table 3 reinforce our previous findings on the lung dataset, demonstrating similarly strong performance when applying the methodology to the BraTS dataset. This cross-domain evaluation further validates the model's robustness and adaptability to diverse data types.

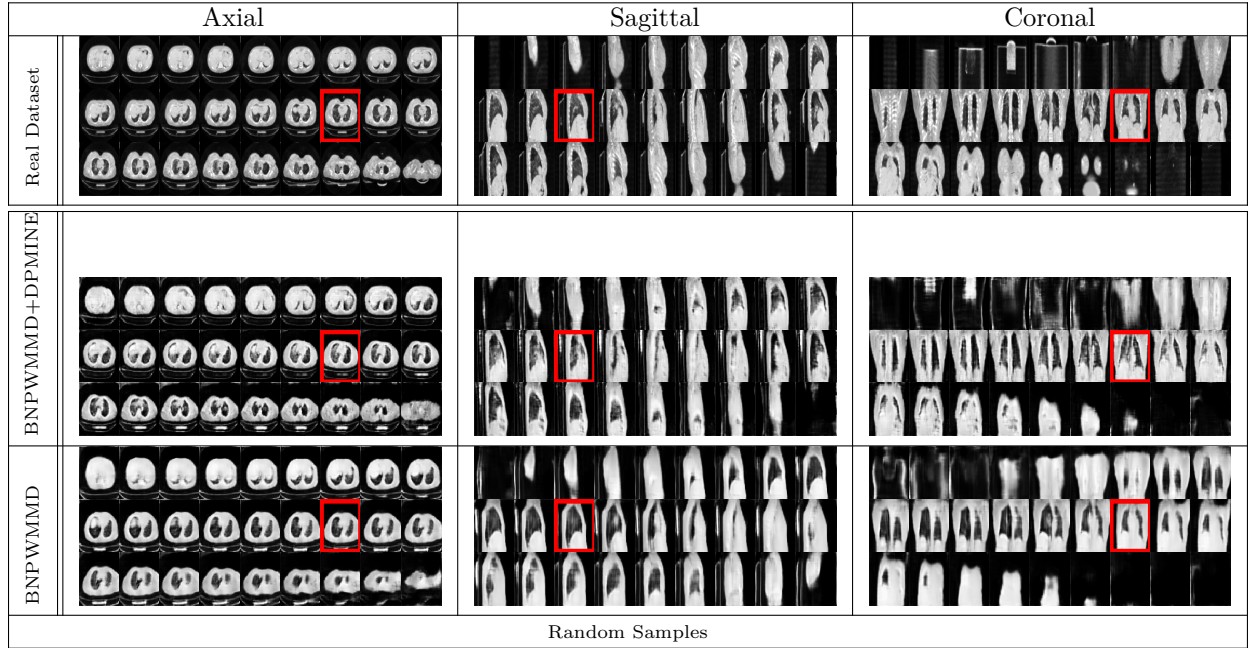

Figure 12: 27 slices along each axis of a 3D sample randomly generated using BNPWMMD+DPMINE after 7500 epochs, compared with BNPWMMD, for the COVID-19 example.

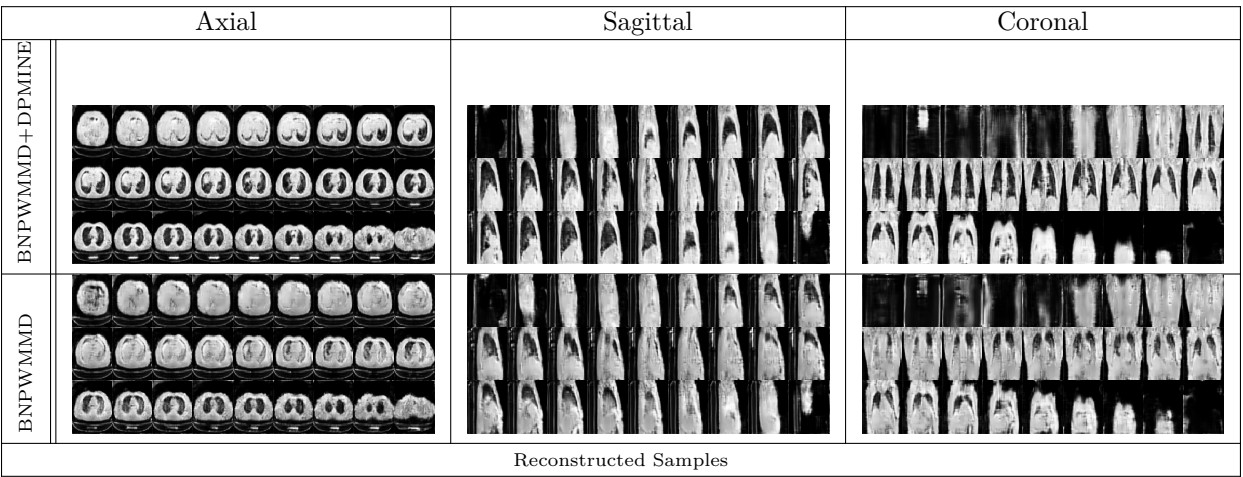

Figure 13: 27 slices along each axis of a 3D sample reconstructed using BNPWMMD+DPMINE after 7500 epochs, compared with BNPWMMD, for the COVID-19 example.

### E.3 Implementing Details

#### E.3.1 Calculation of Evaluation Scores

Considering the real and generated features, $\boldsymbol{f}_r := (\text{Feature1}_{1:200,r}, \text{Feature2}_{1:200,r})$ and $\boldsymbol{f}_g := (\text{Feature1}_{1:200,g}, \text{Feature2}_{1:200,g})$, the FID and KID metrics are calculated using the following Eqs., respectively Bińkowski et al. (2018):

$$\text{FID}(\boldsymbol{f}_r, \boldsymbol{f}_g) = ||\boldsymbol{\mu}_{\boldsymbol{f}_r} - \boldsymbol{\mu}_{\boldsymbol{f}_g}||^2 + \text{Tr}\Big(\Sigma_{\boldsymbol{f}_r} + \Sigma_{\boldsymbol{f}_g} - 2\sqrt{\Sigma_{\boldsymbol{f}_r}\Sigma_{\boldsymbol{f}_g}}\Big), \tag{46}$$

$$\text{KID}(\boldsymbol{f}_r, \boldsymbol{f}_g) = \text{MMD}^2(\boldsymbol{f}_r, \boldsymbol{f}_g), \tag{47}$$

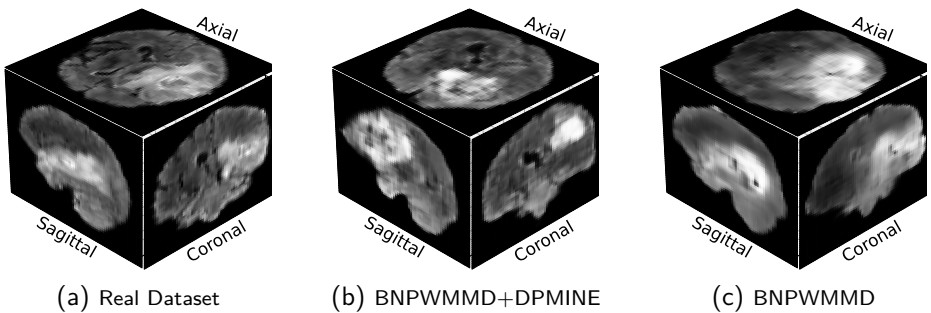

Figure 14: A 3D visualization of a real sample and a randomly generated sample using BN-PWMMD+DPMINE after 7500 epochs, compared with BNPWMMD, in the BraTS18 example.

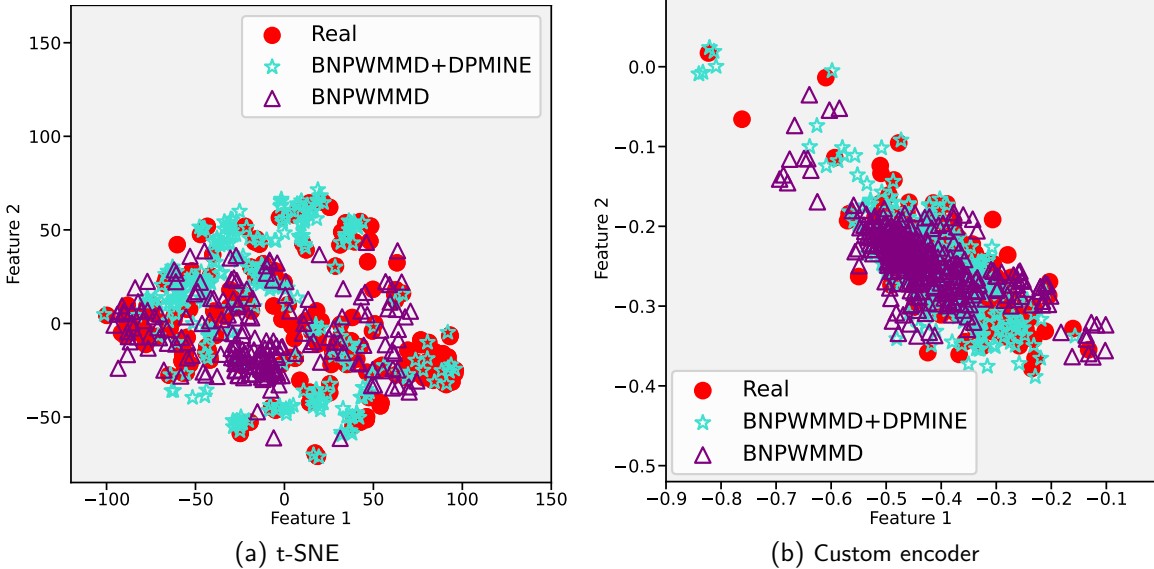

Figure 15: Scatter plots of two-dimensional feature representations in the BraTS18 example, comparing samples generated using BNPWMMD+DPMINE with those generated using BNPWMMD. Panel (a) shows the t-SNE embedding of the features, while panel (b) displays the features obtained from a custom encoder.

where $\boldsymbol{\mu}_{\boldsymbol{f}_r}$ and $\boldsymbol{\mu}_{\boldsymbol{f}_g}$, and $\Sigma_{\boldsymbol{f}_r}$ and $\Sigma_{\boldsymbol{g}_r}$ are the mean vector and covariance matrix of $\boldsymbol{f}_r$ and $\boldsymbol{f}_g$, respectively. The empirical MMD metric in Gretton et al. (2012) is used to compute $\mathrm{MMD}^2(\cdot, \cdot)$, with a polynomial kernel $k(\boldsymbol{f}_r, \boldsymbol{f}_g) = (0.5\boldsymbol{f}_r^{\mathrm{T}}\boldsymbol{f}_g + 1)^{\nu}$ and degree $\nu$. The "Tr" denotes matrix trace, "T" denotes matrix transpose, and "$||\cdot||$" denotes Euclidean norm. We use the provided code at `https://torchmetrics.readthedocs.io/en/v0.8.2/image/kernel_inception_distance.html` to compute KID with $\nu = 3$, and also use the provided code at `https://pytorch.org/ignite/generated/ignite.metrics.FID.html` to calculate FID.

Additionally, we calculate MS-SSIM using the available codes at `https://torchmetrics.readthedocs.io/en/v0.8.2/image/multi_scale_structural_similarity.html`.

### E.3.2 Network Setting

This section summarizes the network architectures used in all experiments. The encoder, generator, and discriminator architectures listed in Tables 4, 6, and 7 follow standard designs commonly adopted in 3D generative modeling and are consistent with those used in the $\alpha$-WGAN setting (Kwon et al., 2019). The BNPWMMD model Fazeli-Asl & Zhang (2023) builds on these components and additionally includes a code

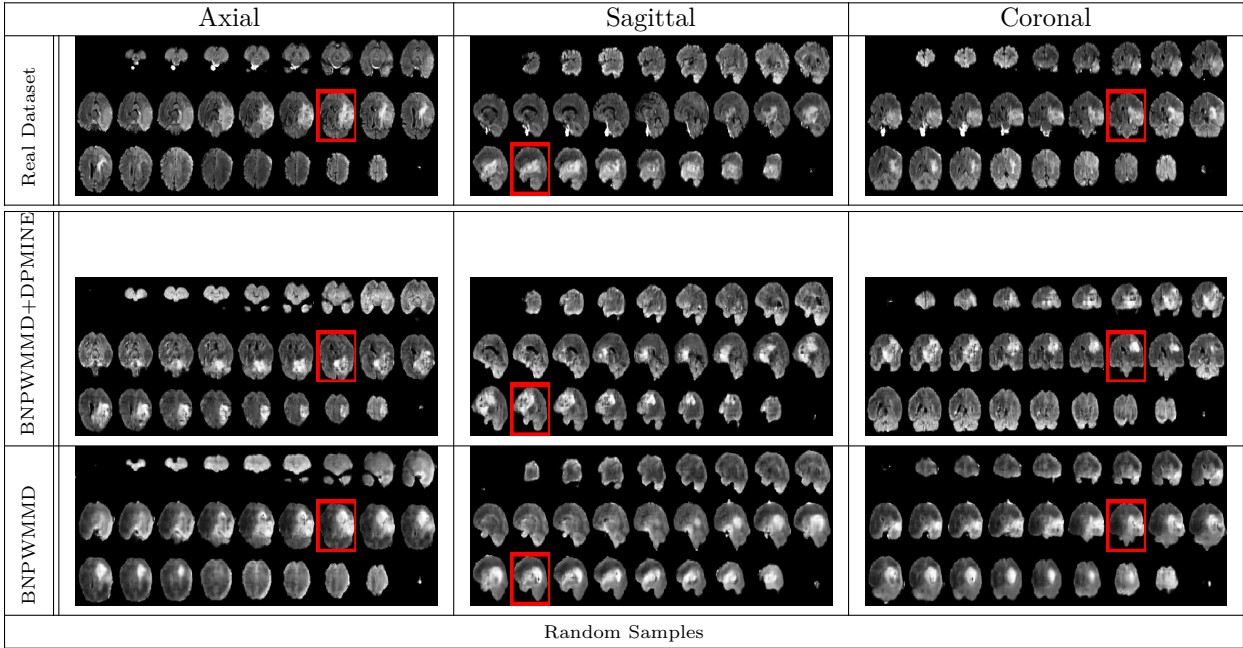

Figure 16: 27 slices along each axis of a 3D sample randomly generated using BNPWMMD+DPMINE after 7500 epochs, compared with BNPWMMD, for the BraTS18 example.

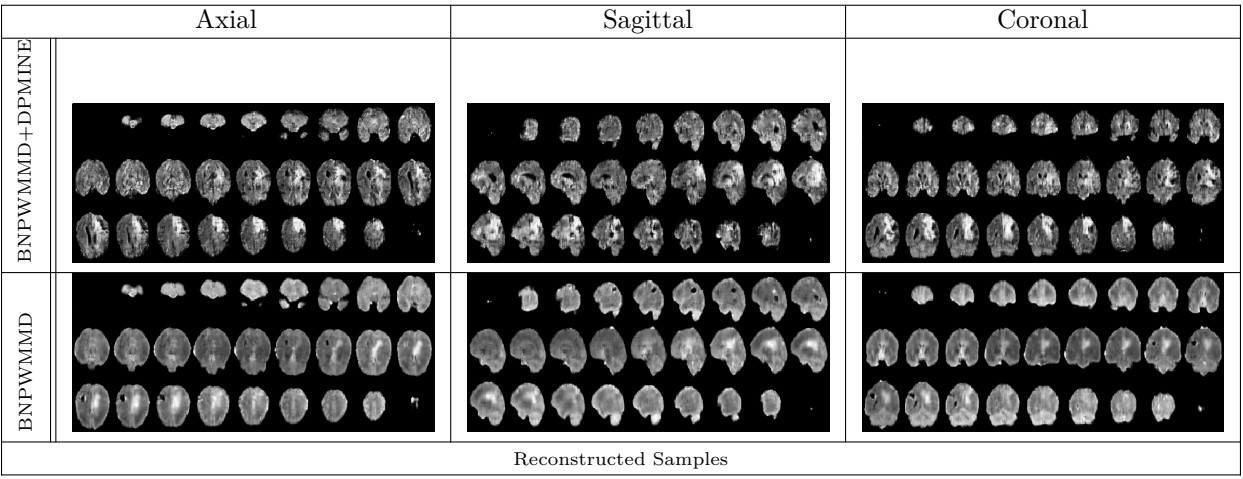

Figure 17: 27 slices along each axis of a 3D sample reconstructed using BNPWMMD+DPMINE after 7500 epochs, compared with BNPWMMD, for the BraTS18 example.

generator, detailed in Table 5. Figure 18 illustrates the overall workflow of the BNPWMMD+DPMINE framework.

Additionally, Table 8 presents a simple architecture used for updating the parameters $\{T_{\boldsymbol{\gamma}}\}_{\boldsymbol{\gamma}\in\boldsymbol{\Gamma}}$ in the DP-MINE and MINE calculationsBelghazi et al. (2018).

All architectures have been implemented in PyTorch using the Adam optimizer and a learning rate of 0.0002 on an NVIDIA Tesla V100-SXM2 with 4 GPUs, each with 32GB of RAM. Our code requires 48-72 hours to provide results for the COVID-19 and BraTS18 examples. For the coil experiments, our code takes about 4-6 hours.

Table 3: Comparison of statistical scores (mean $\pm\,2\sigma$) for the BraTS18 example ver 100 replications.

| Evaluator | | BNPWMMD + DPMINE | BNPWMMD |
|---|---|---|---|
| FID | Custom Encoder | **$0.00197 \pm 0.00015$** | $0.00264 \pm 0.00021$ |
| | t-SNE | **$25.93170 \pm 14.68720$** | $34.13418 \pm 16.00405$ |
| KID | Custom Encoder | **$0.00038 \pm 0.00004$** | $0.00067 \pm 0.00007$ |
| | t-SNE | **$8.81000 \pm 7.18880$** | $9.63544 \pm 8.95555$ |
| MMD | Custom Encoder | **$0.00075 \pm 0.00078$** | $0.00092 \pm 0.00087$ |
| | t-SNE | **$0.19887 \pm 0.06760$** | $0.27312 \pm 0.07383$ |
| MS-SSIM | – | **$0.75007 \pm 0.10722$** | $0.74998 \pm 0.12031$ |

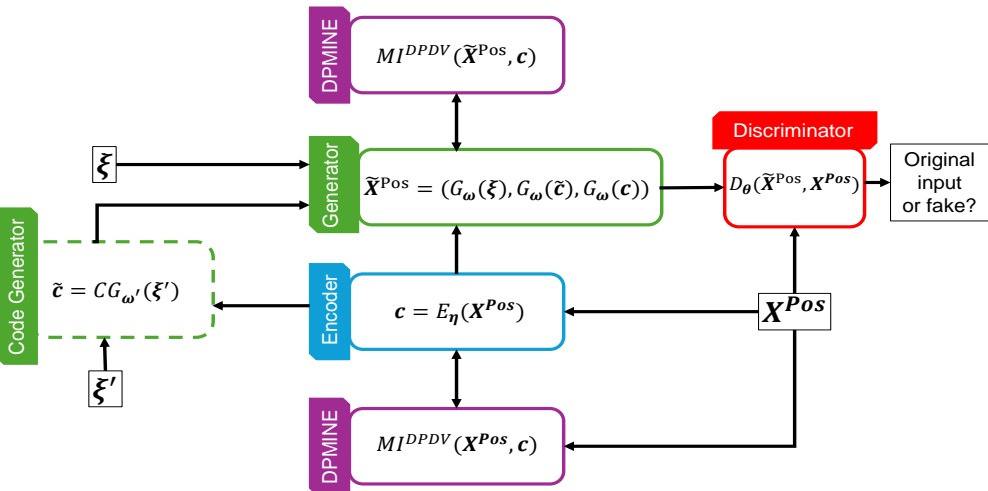

Figure 18: A general flowchart to illustrate the actual operation process of BNPWMMD+DPMINE.

## F  Limitations

The BNP framework proposed in this paper assumes that the training data follows the IID assumption. However, this assumption restricts the model's applicability to certain privacy-preserving techniques, such as federated learning, which often rely on non-IID assumptions at the local device level. While this limitation is acknowledged, it is beyond the scope of the current work to address it. In future research, we plan to extend the proposed BNP framework to incorporate non-IID assumptions and apply it to federated learning. This extension aims to mitigate issues such as slow convergence and unfair predictions that may arise when applying a global model to unseen data.

## G  Broad Impacts and Safeguards

One potential positive impact of this paper is the advancement of artificial intelligence (AI) techniques in healthcare. The proposed model offers a solution to the challenge of accessing high-quality, diverse medical datasets for clinical decision-making. This can lead to improved accuracy and effectiveness in diagnosing and treating diseases, ultimately benefiting patients and healthcare providers. Additionally, the incorporation of a BNP procedure helps uncover underlying structures in the data and reduce overfitting, enhancing the reliability of the generated samples.

However, the use of AI in healthcare also raises concerns about data privacy and security. It is important to prioritize patient privacy and data protection by implementing strict regulations and protocols to ensure that patient data is anonymized and protected. Additionally, there is a risk of bias in the generated samples, which could lead to disparities in healthcare outcomes if the models are not properly validated and tested on diverse populations. Therefore, transparency and accountability in the development and deployment of these AI models are necessary. Rigorous testing and validation on diverse datasets should be conducted to mitigate

Table 4: Encoder: A 3D Convolutional Network Architecture for the COVID-19 Dataset used in BNPWMMD-GAN.

| Layer | Input dimension | Output dimension | Kernel Size | Stride | Padding | Activation Function |
|---|---|---|---|---|---|---|
| Convolution | $1 \times 64 \times 64 \times 64$ (Data dimension) | $64 \times 32 \times 32 \times 32$ | $4 \times 4 \times 4$ | 2 | 1 | Leaky ReLU (negative slope=0.2) |
| Convolution | $64 \times 32 \times 32 \times 32$ | $128 \times 16 \times 16 \times 16$ | $4 \times 4 \times 4$ | 2 | 1 | Leaky ReLU (negative slope=0.2) |
| BatchNorm | $128 \times 16 \times 16 \times 16$ | $128 \times 16 \times 16 \times 16$ | - | - | - | - |
| Convolution | $128 \times 16 \times 16 \times 16$ | $256 \times 8 \times 8 \times 8$ | $4 \times 4 \times 4$ | 2 | 1 | Leaky ReLU (negative slope=0.2) |
| BatchNorm | $256 \times 8 \times 8 \times 8$ | $256 \times 8 \times 8 \times 8$ | - | - | - | - |
| Convolution | $256 \times 8 \times 8 \times 8$ | $512 \times 4 \times 4 \times 4$ | $4 \times 4 \times 4$ | 2 | 1 | Leaky ReLU (negative slope=0.2) |
| BatchNorm | $512 \times 4 \times 4 \times 4$ | $512 \times 4 \times 4 \times 4$ | - | - | - | - |
| Convolution | $512 \times 4 \times 4 \times 4$ | $1000 \times 1 \times 1 \times 1$ | $4 \times 4 \times 4$ | 2 | 1 | - |

Table 5: Code-Generator: A 3D Convolutional Network Architecture for the COVID-19 Dataset used in BNPWMMD-GAN.

| Layer | Input dimension | Output dimension | Kernel Size | Stride | Padding | Activation Function |
|---|---|---|---|---|---|---|
| Convolution | 100 (Sub-latent dimension) | $16 \times 2 \times 5 \times 10$ | $3 \times 3 \times 3$ | 1 | 1 | ReLU |
| BatchNorm2 | $16 \times 2 \times 5 \times 10$ | $16 \times 2 \times 5 \times 10$ | - | - | - | - |
| Max-pooling | $16 \times 2 \times 5 \times 10$ | $16 \times 2 \times 5 \times 5$ | $1 \times 1 \times 2$ | $1 \times 1 \times 2$ | - | - |
| Convolution | $16 \times 2 \times 5 \times 5$ | $32 \times 2 \times 5 \times 5$ | $3 \times 3 \times 3$ | 1 | 1 | ReLU |
| BatchNorm2 | $32 \times 2 \times 5 \times 5$ | $32 \times 2 \times 5 \times 5$ | - | - | - | - |
| Max-pooling | $32 \times 2 \times 5 \times 5$ | $32 \times 2 \times 5 \times 2$ | $1 \times 1 \times 2$ | $1 \times 1 \times 2$ | - | - |
| Convolution | $32 \times 2 \times 5 \times 2$ | $64 \times 2 \times 5 \times 2$ | $3 \times 3 \times 3$ | 1 | 1 | ReLU |
| BatchNorm3 | $64 \times 2 \times 5 \times 2$ | $64 \times 2 \times 5 \times 2$ | - | - | - | - |
| Max-pooling | $64 \times 2 \times 5 \times 2$ | $64 \times 2 \times 5 \times 1$ | $1 \times 1 \times 2$ | $1 \times 1 \times 2$ | - | - |
| Flatten | $64 \times 2 \times 5 \times 1$ | 640 | - | - | - | - |
| Fully-connected | 640 | 1000 | - | - | - | - |

bias and ensure fairness. Collaboration between healthcare professionals, AI researchers, and policymakers is essential to establish guidelines and regulations that prioritize patient privacy, data protection, and equitable healthcare outcomes.

Table 6: Generator: A 3D Convolutional Network Architecture for the COVID-19 Dataset used in BNPWMMD-GAN.

| Layer | Input dimension | Output dimension | Kernel Size | Stride | Padding | Activation Function |
|---|---|---|---|---|---|---|
| Transposed convolution | 1000 (Latent dimension) | $512 \times 4 \times 4 \times 4$ | $4 \times 4 \times 4$ | 1 | 0 | ReLU |
| BatchNorm | $512 \times 4 \times 4 \times 4$ | $512 \times 4 \times 4 \times 4$ | - | - | - | - |
| Upscale | $512 \times 4 \times 4 \times 4$ | $512 \times 4 \times 4 \times 4$ | - | - | - | - |
| Convolution | $512 \times 4 \times 4 \times 4$ | $256 \times 8 \times 8 \times 8$ | $3 \times 3 \times 3$ | 1 | 1 | ReLU |
| BatchNorm | $256 \times 8 \times 8 \times 8$ | $256 \times 8 \times 8 \times 8$ | - | - | - | - |
| Upscale | $256 \times 8 \times 8 \times 8$ | $256 \times 8 \times 8 \times 8$ | - | - | - | - |
| Convolution | $256 \times 8 \times 8 \times 8$ | $128 \times 16 \times 16 \times 16$ | $3 \times 3 \times 3$ | 1 | 1 | ReLU |
| BatchNorm | $128 \times 16 \times 16 \times 16$ | $128 \times 16 \times 16 \times 16$ | - | - | - | - |
| Upscale | $128 \times 16 \times 16 \times 16$ | $128 \times 16 \times 16 \times 16$ | - | - | - | - |
| Convolution | $128 \times 16 \times 16 \times 16$ | $64 \times 32 \times 32 \times 32$ | $3 \times 3 \times 3$ | 1 | 1 | ReLU |
| BatchNorm | $64 \times 32 \times 32 \times 32$ | $64 \times 32 \times 32 \times 32$ | - | - | - | - |
| Upscale | $64 \times 32 \times 32 \times 32$ | $64 \times 32 \times 32 \times 32$ | - | - | - | - |
| Convolution | $64 \times 32 \times 32 \times 32$ | $1 \times 64 \times 64 \times 64$ | $3 \times 3 \times 3$ | 1 | 1 | Tanh |

Table 7: Discriminator: A 3D Convolutional Network Architecture for the COVID-19 Dataset used in BNPWMMD-GAN.

| Layer | Input dimension | Output dimension | Kernel Size | Stride | Padding | Activation Function |
|---|---|---|---|---|---|---|
| Convolution | $1 \times 64 \times 64 \times 64$ (Data dimension) | $64 \times 32 \times 32 \times 32$ | $4 \times 4 \times 4$ | 2 | 1 | Leaky ReLU (negative slope=0.2) |
| Convolution | $64 \times 32 \times 32 \times 32$ | $128 \times 16 \times 16 \times 16$ | $4 \times 4 \times 4$ | 2 | 1 | Leaky ReLU (negative slope=0.2) |
| BatchNorm | $128 \times 16 \times 16 \times 16$ | $128 \times 16 \times 16 \times 16$ | - | - | - | - |
| Convolution | $128 \times 16 \times 16 \times 16$ | $256 \times 8 \times 8 \times 8$ | $4 \times 4 \times 4$ | 2 | 1 | Leaky ReLU (negative slope=0.2) |
| BatchNorm | $256 \times 8 \times 8 \times 8$ | $256 \times 8 \times 8 \times 8$ | - | - | - | - |
| Convolution | $256 \times 8 \times 8 \times 8$ | $512 \times 4 \times 4 \times 4$ | $4 \times 4 \times 4$ | 2 | 1 | Leaky ReLU (negative slope=0.2) |
| BatchNorm | $512 \times 4 \times 4 \times 4$ | $512 \times 4 \times 4 \times 4$ | - | - | - | - |
| Convolution | $512 \times 4 \times 4 \times 4$ | $1 \times 1 \times 1 \times 1$ | $4 \times 4 \times 4$ | 2 | 1 | Sigmoid |

Table 8: Architecture of the shared $T_\gamma$ network employed by DPMINE and all benchmark mutual information estimators.

| Layer | Input Dimension | Output Dimension | Activation Function |
|---|---|---|---|
| concatenation | $1000 + 64^3$ (Latent dimension + Data dimension) | 263144 | - |
| Linear | 263144 | 400 | ReLU |
| Linear | 400 | 400 | ReLU |
| Linear | 400 | 400 | ReLU |
| Linear | 400 | 1 | ReLU |

