# OpenReview forum: "A Bayesian Bootstrap Framework for Mutual Information Neural Estimation: Bridging Classical Mutual Information Learning and Bayesian Nonparametric Learning"
_TMLR — Accepted by TMLR_

### Review · Reviewer_sopd · 2025-11-15

**Summary Of Contributions:**

This paper makes several contributions. First, it introduces a Bayesian nonparametric estimator of mutual information that replaces empirical-distribution–based losses with a loss constructed from a finite Dirichlet process posterior representation. Second, it shows that this regularization reduces variance and mitigates sensitivity to sampling fluctuations and outliers, providing more stable gradients during training and improving convergence of neural MI estimators. Third, it establishes theoretical guarantees for. Fourth, it demonstrates the practical utility of the estimator by applying it to mutual-information maximization in variational autoencoders, yielding improved convergence and representation quality on synthetic data and 3D CT image generation.

The key strengths of the paper are (i) its creative use of BNP ideas, and in particular of prior regularization, to achieve improved and stabilized performance in a DL task such as MINE, and (ii) its experimental validation supporting the claims. Some major weaknesses are the relatively poor organization of the presentation, the lack of clarity in the theoretical proofs, and an insufficient provision of insights about experiments.

**Additional Comments:**

None.

**Audience:**

Yes

**Audience Explanation:**

The contribution fits within a recent interest in the incorporation of Bayesian ideas into modern learning frameworks (e.g., optimization), so a community of readers (though not extremely large) would exist.

**Broader Impact Concerns:**

I do not see any ethical concerns with this work.

**Claims And Evidence:**

Yes

**Claims Explanation:**

The above “yes” is with caveats. The experiments are quite extensive and convincing, although I do believe that their interpretation is a bit hindered by the unclear presentation of the part on GAN (see more later). The theoretical proofs, instead, are extremely unclear, though (as I explain later) I don’t believe the theorems add much to the treatment, so they could either be amended or eliminated altogether.

**Requested Changes:**

Major adjustments:

-	The proofs of the theoretical results are quite messy and, as written, seem to contain flaws. For instance, the proof of Theorem 1 takes an expectation with respect to the posterior law of the sample. However, somehow it treats the $X_\ell$ as constants with respect to the law (as if randomness was only because of the weights J). But the atoms are part of the random measure, and I don’t see a quick way to solve this flaw. I would also encourage the authors to be precise about which quantity ($n$ or $N$) is sent to infinity first, and with respect to which probability measure the a.s. statements should be understood. The proof of Theorem 2 is equally hard to follow. It may be that I am misinterpreting the results and/or the proofs, but I do have some experience with these calculations. This implies that, whether I am misinterpreting or not, the authors should consider improving their treatment of the theoretical results significantly, if they plan to keep those results at all. This is because if someone with some experience is finding the reading of the results hard, I can only imagine how hard it may be for less experienced readers. To be clear, this being a predominantly methodological contribution, I don’t see any obstacle in removing the results altogether, especially if the authors are not able to fix the proofs and/or provide a much clearer presentation. In fact, the results don’t add much to the contribution of the paper, and may only end up creating confusion (at least in their current form).

-	The paper makes many somewhat vague statements, sometimes to simply tie its contribution to existing literature. Moreover, especially in the later part of it, a lot of math is introduced with not enough context. The combination of these two factors makes it somewhat hard to discern the true contributions of the paper from smaller implementation details or weak ties to existing literature. I know this is a bit of a generic request, but I really encourage the authors to think hard about what the essential points of their article are, and to center their writing around those, with a lot focus and parsimony in the presentation. For instance, the transition between the part on MI estimation and the GAN architecture is quite sharp and it is hard to understand the connection until later, and even then that connection is buried behind a wall of equations that are hard to grasp all at once. Please consider revising in a major way the text to take these considerations into account.

Minor adjustments/clarifications:

-	The authors should be careful with typesetting (e.g., there are multiple instances of Eq. equation X, some idiosyncratic indentations), and in general a thorough review of English writing is necessary.

-	Can the authors elaborate further on the MAP strategy put out in Remark 2? My intuition is that one could try to find the optimal $\alpha$ by maximizing the EPPF associated to the DP, but that would lead to choosing an arbitrarily large value in the presence of distinct data points. I am interested in knowing more about this procedure.

-	In the simulated data experiments (e.g., figures 2 and 3), how does hyperparameter (say, DP concentration and centering measure) choice affect the results? What I mean is: is the DP very concentrated on a good guess for the data-generating process? If so, the result is expected due to prior regularization. This is not bad in itself, but I guess the choice of hyperparameters concentrating the prior on a good guess need to be justified by actual prior (e.g. scientific) beliefs about the data-generating process, not just (as it may be in this case) by the knowledge of the authors of the generative process. Summarizing, the main thing I’m asking is to add intuition as to what features of the proposed methodology lead to better performance (in this case, lower variability), and a discussion as to how the choices leading to these features are justifiable in practice.

---

> ### Author Response · Authors · 2025-12-17
> **Response to Reviewer sopd**
>
> We sincerely thank the reviewer for the careful reading of the manuscript and for the thoughtful, constructive comments, which have helped us significantly clarify the theory, sharpen the focus on the main contribution, and improve the overall presentation of the paper.
>
> **Requested Changes:**
>
> **1) Theoretical proofs**
>
> - Regarding Theorem 1, the randomness arises from both the weights and the atoms, and the expectation must account for both sources. We have corrected the proof by applying the law of total expectation and now provide full details to clarify this point. We have to clarify that the theorem is presented by letting $n\rightarrow\infty$, for any $N>0$, which is now stated in the updated version.
>
> - Theorem 2 has been removed from the paper, as noted by the reviewer and Reviewer wpM7, since it did not add substantial value to the contribution.
>
> **2) Focus on essential point of the paper (main contribution)**
>
> - We have substantially revised the paper, as described in our responses to the other reviewers, to clearly center the presentation on the main contribution. The text has been reorganized to improve coherence and to better distinguish the core ideas from implementation details and secondary connections to existing literature. In addition, the mathematical exposition in the main paper has been simplified, with non-essential derivations removed or deferred, so that readers can more easily follow the central contribution without being overwhelmed by technical details.
>
> **3) MAP strategy**
>
> - We thank the reviewer for this question. Remark 2 has been removed, as it contained non-essential material, and this removal was also suggested by Reviewer wpM7.
>
> - The MAP strategy used in the experimental section follows the numerical procedure proposed in [Fazeli-Asl & Zhang (2023, Algorithm 1)](https://arxiv.org/pdf/2308.14048). In that work, the strategy maximizes $F^{\mathrm{Pos}}$ with respect to the concentration parameter $a$. The theoretical motivation and details of this approach are provided in [Fazeli-Asl & Zhang (2023, Section 4.2)](https://arxiv.org/pdf/2308.14048), where the behavior of this optimization is carefully analyzed. We now explicitly cite this section in the revised paper to guide interested readers who seek further clarification of the procedure.
>
> **4) Isolating DP hyperparameters**
>
> We thank the reviewer for raising this important question. We have addressed this issue through newly added numerical results (Figure 3), as also discussed in our response to Requested Change #6 of Reviewer 77Se.
>
> The results illustrate that when the centering measure $H$ provides a reasonable prior guess for the data-generating mechanism, increasing the concentration parameter $a$ has little effect on performance, reflecting a standard prior regularization effect. In contrast, when $H$ is poorly specified, increasing $a$ can bias the posterior bootstrap samples toward the prior, while small values of $a$ yield more stable behavior. This aligns with classical intuition in Bayesian nonparametrics, where the concentration parameter controls the strength of prior regularization relative to the data.
>
> We also observe that the MAP-based strategy consistently yields good performance across different settings. Further discussion is provided in Section 4.1.3 of the revised paper.

---

> > ### Comment · Reviewer_sopd · 2026-01-03
> >
> > I thank the authors for their edits and responses. The paper is now more readable and accessible; however, I strongly recommend one further round of revisions focusing on language and overall presentation.
> >
> > One final comment is that, while it is appreciated that the authors accepted the reviewers’ invitation to remove unnecessary references, I believe that some previously included references situating this work within the BNP learning literature have been incorrectly removed. I encourage the authors to appropriately cite prior work that employs the fundamental idea of modeling the data distribution via Bayesian priors (e.g., within the robust learning framework), as I believe this constitutes the primary community of reference for the present article.

---

> ### Author Response · Authors · 2026-01-04
> **Response to Reviewer sopd**
>
> We sincerely thank the reviewer for bringing this to our attention. Among the most closely related Bayesian nonparametric works to our paper, the study by Dellaporta et al. (2022) was the only relevant reference that was inadvertently missed, as the works of Fong and Lyddon were already discussed in Section 3.2. This oversight has now been corrected, and the paper by Dellaporta et al. has been added and discussed in the last two paragraphs of Section 3.2 in the revised manuscript.
>
> Regarding the overall presentation, we have made substantial revisions in response to the reviewers’ comments. In particular, the Introduction and Abstract have been rewritten to clearly emphasize the main objective of the paper, namely the development of a bootstrap-based MINE framework, while avoiding an emphasis on generative modeling procedures.
>
> As suggested by two reviewers, we have added a dedicated background section reviewing the Bayesian bootstrap framework of Fong and Lyddon, and we now explicitly position our method as a posterior bootstrap procedure within this framework. In addition, the experimental section has been reorganized to better reflect the main goals and conclusions of the paper. We designed more challenging experiments to evaluate the performance of the proposed MINE bootstrap method and to systematically examine its sensitivity to the choice of Dirichlet process hyperparameters.
>
> Overall, the manuscript has been carefully reorganized and rewritten step by step based on our understanding of all reviewers’ suggestions. We will also perform a final pass to make minor improvements in writing and clarity.

---

### Review · Reviewer_77Se · 2025-11-26

**Summary Of Contributions:**

This paper proposes DPMINE, a Bayesian nonparametric method for the estimation of mutual information (MI). This is done through placing a Dirichlet Process (DP) prior over the unknown joint data-generating distribution and propagating the posterior uncertainty via a neural MI estimator. The method draws approximated DP posterior samples, through a finite approximation of the DP, generates synthetic data from each sample, and uses a MINE-style variational objective (DV or JS) to estimate the MI for each posterior draw.  A further contribution, described in Section 3.3, is the proposed integration of DPMINE into deep generative models, where the MI posterior or its summary statistics, such as posterior mean, is used as a regularizer or objective in representation learning.

**Audience:**

Yes

**Audience Explanation:**

Mutual information estimation is central to many problems in machine learning, including representation learning, generative modelling, and causal inference. Providing a Bayesian quantification of uncertainty over MI is appealing, and the attempt to integrate such an estimator into deep generative models is timely. The conceptual combination of DP-based posterior sampling and neural MI estimators is novel, and the ambition of producing a posterior over MI is likely to attract interest. However, I believe a clearer presentation of the methodology and results , as well as a thorough discussion of the issues raised above would lead to a stronger theoretical grounding and a clearer characterisation of limitations, making the work more accessible to a larger audience.

**Broader Impact Concerns:**

No broader impact concerns

**Claims And Evidence:**

No

**Claims Explanation:**

I think the proposed estimator is appealing, the idea is novel, and the method shows strong empirical performance. However, I believe that in its current form, the manuscript lacks adequate justification of some methodological and theoretical aspects. Moreover, I think a clearer and more structured presentation of the background and methodology would substantially improve the accessibility of the work and better motivate the contributions. I outline these concerns in detail below:

- The method appears to rely on a form of bootstrapping that is closely related to the BNP techniques discussed in the paper. However, the presentation does not clearly explain how it connects to this existing body of work, nor does it fully clarify the methodology itself. In particular, the proposed procedure seems to involve:
    - Sampling (approximately) probability measures from the DP posterior.
    - For each sample, defining an estimator of the mutual information by maximising a VLB that is constructed using that posterior draw.
    - Treating the resulting collection of mutual information estimators as posterior samples of the MI functional.

     This is effectively the posterior bootstrap, which is closely related to the Bayesian bootstrap, as developed by Lyddon (2018), Fong (2019), with the main distinction being that the functional of interest here is the MI estimator rather than a parameter minimising a loss. Although these papers are cited, the relationship between the proposed approach and these bootstrap methodologies is not discussed in sufficient depth.

- The discussion of the Dirichlet Process (DP) as a nonparametric Bayesian model is, at times, conflated with the broader BNP perspective on belief updating. For example, Section 3.1 is devoted to the DP and its approximation, yet much of the accompanying discussion, such as Remark 2, which addresses prior elicitation, relies on arguments drawn from papers situated within the Bayesian Nonparametric Learning (BNPL) framework. Because the BNPL framework has not been introduced at that point in the paper, it becomes difficult for the reader to follow the motivation and reasoning behind these arguments. A similar issue appears in Remark 1, where the authors claim that using a prior centering measure in a DP “makes the model less sensitive to fluctuations […] for optimization processes in models that rely on loss functions.” It is unclear which model or loss function this statement refers to, leaving the reader without sufficient context. Furthermore, the work by Bariletto (2024) cited here is grounded in a Distributionally Robust Optimization framework, in which the role and interpretation of the centering measure differ substantially from its role in standard Bayesian nonparametric methods.

- Some claims in the paper, while potentially correct, are not sufficiently justified or explained. For instance, in multiple places (e.g. the beginning of Section 3 and in Section 1), the authors state that the proposed method is “more robust” than existing approaches. However, it is not clear what form of robustness this refers to until quite a bit later in the paper. This clarification is crucial, especially because the proposed approach differs in some ways from standard BNP methodologies, which, as the authors themselves note, typically address robustness to model misspecification in parametric inference. Hence, a clear explanation and position of the method within the literature would add to the overall motivation of the method.

- I think Theorems 1 and 2 are insightful. However, a few theoretical issues are not discussed:
  -  Theorem 2 states that, conditionally on a posterior sample, the neural MI estimator converges to the true MI of that sampled distribution as the amount of synthetic data grows. However, it is not entirely clear to me if this would imply that the posterior distribution of MI converges to the true MI. This concern is amplified by the fact that the functions $f_i$, which produce joint and product-of-marginals samples, are not properly defined, creating ambiguity in the interpretation of the theorem
- The approximated DP posterior draws are random discrete probability measures, whereas the true MI is (presumably) defined for an underlying continuous distribution. Even if DPMINE computes the DV bound exactly for a discrete DP draw, would this necessarily imply that it provides a good approximation to the MI of the continuous data-generating distribution? This distinction is not adequately discussed in the paper.
- How does the choice of truncation of the DP introduce any bias or error in the MI estimation

- The presentation of the methodology and the results are unclear in places. For example, theorem 1 mentions equation 15, which has not yet been introduced. The functions $f_i$ are not properly defined or explained in the context of the method.

- I found that discussions around potential limitations of the method and empirical results about how hyperparameters like $\alpha$ or $N$ could affect the method are not present, although they could enhance the clarity of the paper.

**Requested Changes:**

I believe this paper is a potentially valuable contribution to TMLR. Strengthening the clarity of the exposition and addressing some methodological and theoretical gaps would substantially improve the manuscript and make its contributions more accessible and compelling. Hence, I propose the following 'critical' changes:
- A detailed step-by-step description of the proposed algorithm, together with an explicit discussion about any connection to existing BNP algorithms such as the posterior bootstrap and Bayesian bootstrap frameworks, would help situate the work and clarify its novelty.
- In the Background, could the authors clarify and separate the discussion around the DP, as a nonparametric model for the data-generating process, and the general nonparametric learning framework of Lyddon, Fong as a bootstrap sampling algorithm.
- The paper should clearly define and analyse the sampling operators $f_1$ and $f_2$, explaining how synthetic joint and product-of-marginals samples are produced, and discuss how these are or aren't incorporated into the theoretical results.
- Could the authors elaborate more on the type of robustness induced by the methodology earlier on in the manuscript, providing some intuition about this robustness?
- The authors should address the mismatch, if any, between discrete DP posterior draws and the continuous MI functional (more details in the answer above)
- Could the authors discuss whether the BNP MINE is supposed to yield (non-standard) posterior beliefs about the MI estimator, similarly to previous Bayesian nonparametric learning works? If so, could they clarify the expected behaviour of this posterior, beyond the consistency result provided for a single posterior draw? Is this induced posterior expected to converge to the true MI? If a formal theoretical statement is not possible, a discussion around why that is the case and some intuition behind this behaviour would still be very valuable.
- [non-critical] Could the authors discuss empirically or otherwise the effect of a 'sub-optimal' choice of a prior centering measure $H$, the selection of the hyper-parameter $\alpha$ ? It seems to me that all of these could potentially affect performance, for example, through a large value of $\alpha$ when a bad choice of $H$ is made. This is to be expected in any nonparametric learning method and doesn't necessarily undermine the method's validity, but a discussion (beyond that of Remark 2) or toy example showing for example, how DPMINE’s performance varies with $\alpha$ in the generative-model setting, would strengthen the paper.

---

> ### Author Response · Authors · 2025-12-17
> **Response to Reviewer 77Se**
>
> We would like to extend our sincere thanks to the reviewer for the detailed guidance on the Bayesian bootstrap framework, which enabled us to reorganize the paper during the rebuttal period and significantly strengthen the presentation.
>
> **Requested Changes:**
>
> **1) Explicit discussion on the Bayesian bootstrap framework**
>
> - The abstract and introduction now clearly emphasize the main contribution, namely a Bayesian bootstrap formulation of MINE. Section 3 has been reorganized to provide the necessary background on Bayesian nonparametric learning and to explicitly connect the proposed method to existing Bayesian and posterior bootstrap frameworks. In addition, a step-by-step posterior bootstrap algorithm has been added to clearly describe the proposed approach and highlight its novelty.
>
> - To address the request for a separate discussion of the DP and BNPL frameworks (Lyddon & Fong), a new Subsection 3.2 has been added.
>
> **2) Operators $f_1$ and $f_2$**
>
> - We have stated that $f_1$ and $f_2$ can be any continuous functions, and their specific forms are now clarified in each application.
>
> - In the downstream task using DPMINE within BNPWMMD-GAN, the roles of these functions are explicitly defined in Equations (20) and (21). In Equation (20), $f_1(\cdot)$ is the identity function and $f_2(\cdot) = E_{\boldsymbol{\eta}}(\cdot)$ is the encoder. In Equation (21), $f_1(\cdot) = E_{\boldsymbol{\eta}}(\cdot)$ and $f_2(\cdot) = (G_{\boldsymbol{\omega}} \circ E_{\boldsymbol{\eta}})(\cdot)$, which is the composition of the generator and the encoder.
>
> - In the experimental results section (Section 4.1), the functions are also clarified. For $X = (U, V)$, we use $f_1(X) = U$ and $f_2(X) = V$, which correspond to simple projection functions.
>
> - We also clarify that joint and marginal distributions are not computed explicitly. The auxiliary network $T_{\boldsymbol{\gamma}}$ is trained to distinguish between samples from the joint distribution and samples from the marginals, as already stated. Joint samples are provided directly, while marginal samples are obtained using a standard permutation trick, which is common in neural-based mutual information estimation.
>
> **3) Type of robustness**
>
> - We agree with the reviewer that the term *robustness* may be confusing, as it is commonly used in the BNP literature to refer to robustness to model misspecification. In our paper, the term was intended only to describe the increased stability of the proposed approximation due to its lower variance. To avoid confusion, we no longer use the term *robustness* in this context and instead explicitly refer to this property as **stability**, emphasizing its connection to controlled variance reduction in the proposed procedure.
>
> **4) Discreetness of DP**
> - We now explicitly discuss the discreteness of the Dirichlet process in Sections 3.1 and 3.2 and explain how this property enables an approximate posterior on the parameter of interest through the posterior bootstrap. As in previous Bayesian nonparametric learning works, the discrete DP prior is used to construct posterior bootstraps. This can be viewed as analogous to using Riemann sums to approximate integrals, serving as an intuitive illustration of the approximation mechanism.
>
> **5) Posterior beliefs about the MI**
> - We thank the reviewer for this insightful question. The proposed BNP MINE is designed to yield posterior beliefs over the MINE estimator. This point is now clarified in the paper, and the corresponding posterior bootstrap algorithm is provided.
>
> -  We also include numerical results that display posterior bootstrap samples of MINE; all figures (1-5,9-10) now include the posterior mean to illustrate the behavior of the induced posterior and show that it remains close to the true mutual information.
>
> - Finally, Theorem 2 has been removed, as it played a non-essential role in the paper and this change was also suggested by other reviewers.
>
> **6) Isolating DP hyperparameters**
>
> - We thank the reviewer for this suggestion. We have added numerical results based on toy examples to empirically illustrate the effect of varying the Dirichlet process hyperparameters $(a, H)$; see Figure 3. The results show that when the centering measure $H$ is poorly specified, increasing the concentration parameter $a$ can cause the posterior bootstrap samples of the mutual information to concentrate around the value implied by $H$. In contrast, for small values of $a$, the results are largely invariant to the choice of $H$. Further discussion and analysis are provided in Section 4.1.3 of the revised paper.

---

> > ### Comment · Reviewer_77Se · 2026-01-12
> >
> > I thank the authors for the edits and responses to all reviews. I think the paper has significantly improved and is now more accessible. However, I do agree with reviewer sopd that some further edits on the readability and clarity are needed. Specifically:
> >
> > - In the new section, sometimes (e.g. before equation 14) the term 'push-forward measure' is used to describe a functional estimator of $\gamma$ which is not a measure. Also, functionals like $\hat{\gamma}$ and $\psi$ are not properly defined, indicating the input and output space which hinders readability. A careful edit of this notation would be useful for the readers.
> >
> > - I appreciate that the DPs section and the BNPL section are now separate, and the paper is positioned more clearly within the literature. However, the posterior bootstrap algorithm and MI posterior estimator in Equation 16 seem somewhat disjoint from the rest of the paper. In particular, equation (16) shows that the MI estimator is defined with respect to $\hat{\gamma}(F^{pos})$, which is a random sample from the posterior bootstrap algorithm. However, in Theorem 1, this randomness is not taken into account explicitly as $\gamma$ is taken to be constant, and randomness is induced only with respect to the nonparametric posterior for the data process. Could the authors clarify the connection between the two?
> >
> > - Also, just to clarify and apologies if my previous comment was unclear, given other discussions, my original comment about misaligned references referred to cited papers that were presented as being part of a different literature or not correctly explaining their connection to the current paper (i.e. BNPL and robust learning papers used to explain prior elicitation in Bayesian nonparametrics in the DP section). My suggestion was to carefully explain how all the cited work relates to this current paper, rather than the removal of citations. I believe another careful edit of the literature and citations would be beneficial, and clear positioning of existing work would only make the paper more accessible to interested audiences.

---

> ### Author Response · Authors · 2026-01-29
> **Response to Reviewer 77Se**
>
> We appreciate the reviewer’s continued engagement and the comments provided. We respond to each point below.
>
> **Comment 1**
>
> - We have revised the section to improve precision and readability. In particular, we removed the use of the term “push-forward measure” in places where it was incorrectly used to describe a functional estimator rather than a measure.
>
> - We also clarified the definitions of functionals such as $\hat{\gamma}$ and $\psi$ by explicitly stating their input and output spaces whenever they are introduced.
>
> **Comment 2**
>
>
> - Throughout the paper, we have revised the notation to make this distinction explicit. In particular, in Theorem 1 we now use $\hat{\boldsymbol{\gamma}}$ and $\dot{\boldsymbol{\gamma}}$ on the left- and right-hand sides of the equations, respectively, to clearly separate the estimated parameter from its population counterpart.
>
> - Regarding the treatment of the randomness in $\hat{\boldsymbol{\gamma}}$, the apparent discrepancy arises because $\hat{\boldsymbol{\gamma}}$ is obtained from a draw of the DP posterior. In Theorem 1, the expectation is taken with respect to the nonparametric posterior itself, so this randomness is automatically averaged out. Equivalently, $\hat{\boldsymbol{\gamma}}$ is treated as fixed conditional on a posterior draw, and the theorem characterizes the resulting average behavior. This treatment is standard in posterior bootstrap-based Bayesian nonparametric learning, and is consistent with related results, for example Theorem 3 of Dellaporta et al. (2022). We have provided this clarification after Theorem 1.
>
> **Comment 3**
>
> We have carefully revised the literature discussion to better explain how the cited works relate to the current paper, particularly in the context of Bayesian nonparametric learning. In this revision, we added several previously missing references, including Lyddon et al. (2019), Dellaporta et al. (2022), and Fazeli-Asl et al. (2024), and clarified their roles in motivating our approach. These changes improve the positioning of the paper within the existing literature and make the connections between prior work and our contributions clearer.

---

### Review · Reviewer_wpM7 · 2025-12-02

**Summary Of Contributions:**

The core idea of the paper is to apply Bayesian bootstrap resampling to minibatches used in MINE, replacing uniform sample weights with Dirichlet-resampled weights. This is a well-known technique (Rubin, 1981) but is here repackaged as a “Dirichlet-process MINE” and applied in the context of VAE-GANs. Concretely, the method is inserted into MMD-GAN training using Gaussian kernels, along with an encoder-decoder structure closely resembling MMD-GAN (Li et al., 2017). The contribution appears to be a bit incremental: MMD-GAN + bootstrap-weighted MINE. This would not have been a core issue if the contribution was supported by convincing quantitative results.

**Audience:**

Yes

**Audience Explanation:**

Mutual information estimation is a hard and important problem in ML and statistics, and Bayesian bootstrap is an interesting addition to standard MMD-GAN losses.

**Claims And Evidence:**

No

**Claims Explanation:**

- The paper suffers from a bit of an identity mismatch: it presents itself as proposing a new mutual information estimator, but it evaluates only against MINE, a rather outdated and unstable baseline. Comparisons to modern MI estimators such as Nested Monte Carlo (NMC, Anastasiou et al., McAllester, Rainforth, etc.), Variational Prior Contrastive Estimators (VPCE), InfoNCE/CPC, NWJ, and other $f$-divergence–based methods that directly address variance, bias, and stability issues, would be mandatory in this case. As a result, it is unclear whether the contribution is meant to be a new MI estimator, a robustness trick for MINE, or simply a Bayesian bootstrap trick to improve MMD-GAN. Clarifying this intended role and including comparisons to the appropriate state-of-the-art MI estimators would substantially strengthen the paper, help position the work properly within the literature, and let readers to judge whether the DP-based weighting provides notable gains.

-  The writing needs improvement. The **Abstract** and **Introduction** can be tightened a lot to get to the gist of the method faster. Currently, it reads like a lot of jargon-heavy AI-generated text that heavily dilutes the contributions (which I suggest should be clearly stated at the end of Introduction, instead of the redundant enumeration of the following sections).

- On a related note, the paper really needs to focus its contribution. The subsections in Section 2 are not well connected, contain lots of generic statements and surface summaries of references. Is the goal of the paper to validate a new MI estimator or to improve VAE-GANs? I suggest the authors streamline the readability of Section 2 using paragraphs containing a *single idea that flows into the next one*, and create a dedicated **Related Work** section which discusses direct contender methods (those that are later compared against the proposed metod).

- Overall, I found the notation unnecessarily laborious. Defining everything in terms of CDFs instead of PDFs makes sense for general RVs, but the authors never consider discrete variables or measure-theoretic notation, so I think it can be simplified in terms of PDFs. That way, the notation can also connect to the related work on GANs. I did not see any benefit of having triple subscripts in **Section 3** and the math exposition is at times inconsistent. Furthermore, Theorem numbering in the Appendix does not correspond to Theorem numbering in the main text (e.g., Theorem 3 is Theorem 1, and so on).

- Moreover, I am afraid that the proof of Theorem 2 is circular, as it assumes the function approximator class is rich enough to contain or uniformly approximate the DV minimizer and the network training can reach the optimum. The authors are basically assuming that because the DV bound converges and max is continuous, everything works . Because of this, I find the formalism of the “Theorems” (together with the "Remarks") to be, if anything, more discombobulating than enriching.

- Finally, the evaluation is lacking in many respects, as Table 2 contains the only quantitative benchmark that clearly shows the benefit of the method over naive MINE, but these are > 7 year old contenders. It is also missing consistent decimal formatting and error bars.

**Requested Changes:**

- The contributions need to be focused, the positioning of the paper clarified, the contenders of the method clearly stated and compared against, the writing needs to be streamlined.

- I would suggest that all equations should be numbered and use consistent symbols (e.g., L vs calligraphic L). The remarks before the theorems are unnecessary and pretentiously formal. The theorems are semi-formal and rather argumentative, questioning the need for having them or emphasizing them too much. The entire mathematical exposition can be greatly simplified. Please ensure all typos and inconsistent equation references are fixed (e.g., already the first equation is referenced as "Eq. equation 1", and this applies to all further equations).

- The proposed method should be evaluated with proper quantitative benchmarks and the presentation of the results needs to be considerably improved; the current quality of the figures (squished, sheered, and accompanied by uninformative captions) is unacceptable by any standard.

---

> ### Author Response · Authors · 2025-12-17
> **Response to Reviewer wpM7**
>
> We sincerely appreciate the reviewer’s detailed and constructive feedback, which has significantly strengthened the paper and clarified its positioning. We respond to each comment below.
>
> **Requested Changes:**
>
> **1) Focus, positioning, and related work**
>
> - The abstract and introduction have been fully rewritten to clearly state the paper’s main contribution: a Bayesian bootstrap framework for Mutual Information Neural Estimation (MINE) within Bayesian nonparametric learning. All sections have been revised to align with this focus. Section 3 now introduces background on Dirichlet processes and posterior bootstrap (new Section 3.2), extends the framework to MINE in Section 3.3, and demonstrates its application to a downstream generative modeling task in Section 3.4.
>
> - The title has been revised to more clearly reflect the paper’s main contribution.
>
> - Section 2 has been restructured as a **Related Work** section and rewritten into clear, logically flowing paragraphs. It now discusses directly relevant and competing methods, including those later used for comparison with the proposed approach.
>
> **2) Mathematical clarity and notation**
>
> - All equations are now numbered, notation is consistent (e.g., using $L$ instead of $\mathcal{L}$), and typos and equation reference errors have been fixed. The mathematical exposition has been simplified; remarks before the theorems were removed, Theorem 2 was deleted, and Theorem 1 is retained to present the key theoretical result on the tighter lower bound from the posterior mean of the posterior bootstrap MINE.
>
> - We thank the reviewer for the suggestion to change the CDF notation to a PDF formulation; however, we respectfully retain the use of CDF notation to remain consistent with the Bayesian nonparametric literature. In this work, the underlying data-generating distribution is treated as a random probability measure with a prior placed directly on this measure, and the randomized objective is formulated as a functional of the data distribution. Since the analysis is primarily measure-based rather than density-based, the CDF notation is more natural and is therefore kept throughout the paper.
>
>
> **3) Proper quantitative benchmarks**
> - The form of all figures has been reorganized to improve the clarity and quality of the presented results.
>
> - We have added InfoNCE/CPC, NWJ, and JSD-LB ([Dorent et al., 2025, the most recent state-of-the-art benchmark](https://openreview.net/pdf?id=1gCUv4SzaZ#page=5.71)) to the paper and provided new experimental results for these methods.
>
> - We also respectfully clarify why VNMC and VPCE are not included as benchmarks. As noted in [Ivanova et al., 2024 (Data-Efficient Variational Mutual Information Estimation via Bayesian Self-Consistency)](https://openreview.net/pdf?id=QfiyElaO1f#page=6.09), VNMC and VPCE require at least one closed-form distribution and estimate mutual information through nested Monte Carlo procedures based on Bayes’ rule. This setting is fundamentally different from distribution-free neural estimators, such as InfoNCE/CPC, NWJ, JSD-LB, and the proposed method, which rely only on samples and make no analytic assumptions about the underlying distributions. As a result, direct numerical comparisons between these two families are not meaningful. These distinctions, along with references to VNMC and VPCE, are now clearly discussed in the Related Work section.
>
> - Also, since the main contribution of this paper is a Bayesian bootstrap framework for MINE, the downstream task focuses on applying DPMINE within BNPWMMD-GAN. Accordingly, we compare only BNPWMMD-GAN with its DPMINE-enhanced version. As the paper does not introduce a new generative model, comparisons across multiple generative models are not meaningful. For this reason, experimental results involving other generative models from the previous version have been removed, as they were not aligned with the paper’s main contribution.

---

> > ### Comment · Reviewer_wpM7 · 2026-01-13
> >
> > I appreciate the author's thorough review, which has resulted in an almost new (and in my eyes, improved) paper. Besides R77Se points and a thorough proofreading, I have a few minor remaining comments:
> >
> > - The **Related Work** section currently has one enormous paragraph and one mini-paragraph. Consider separating it into smaller paragraphs, each dedicated to a family of methods. Also, there is no need to write that comparisons are "not meaningful" for certain families of methods, if these are properly categorized as distinct from MINE. This is also in view of the fact that claims "X is not a direct competitor of Y" appear a bit too strong; simply being non-parametric doesn't make a method incomparable to parametric methods (which can be very flexible using universal density approximators). I would also not call most of these methods "classical".
> > - **Section 3.1** ends abruptly. Perhaps something went missing. The last two paragraphs appear incomplete or are written in a way that is extremely hard to follow.
> > - **Algorithm 1** is not helpful and has missing terms (e.g., the variable over which we minimize in line [no line numbers] is missing from the RHS, $H$ becomes $H^*$,...). Ideally, the inputs are explained in plain English.
> > - For my taste, some parts of the paper are still unnecessarily full of rigorous measure-theoretic lingo that harms readability and does more to obfuscate than clarify. This is more than a prompt to streamline than a must-request. Some notation can be drastically simplified (e.g., having a subscript with its own superscript that has its own superscript (e.g., Eq. 16) definitely calls for a change). I have seldomly seen a more involved description of good old (VAE)-GANs than the one in 3.4.
> > - The empirical evaluation has improved/ I would probably refrain from calling the reconstruction results on the COVID-19 dataset "excellent" given the clearly visible distortions in both generated samples (Figure 7). It is enough to show that the DPMINE estimator brings about a measurable benefit (which it seems it does).

---

> ### Author Response · Authors · 2026-01-28
> **Response to Reviewer wpM7**
>
> We thank the reviewer again for the valuable comments and suggestions. We address each point below.
>
> **1) Related Work**
>
> - We have revised the Related Work section by breaking it into several shorter paragraphs, each focusing on a different family of methods.
>
> - We removed statements saying that certain comparisons are “not meaningful” and instead describe these methods as distinct from MINE-based approaches without making strong claims about comparability.
>
> - We also avoided wording such as “not a direct competitor” and replaced the term “classical” with “standard” throughout the section. These changes were made to improve clarity and to better reflect the flexibility of existing methods.
>
>
> **2) Section 3.1**
>
> - We have carefully revised Section 3.1 to address the issue. In particular, we corrected equation (8) by adding the missing term $J^{\mathrm{Pos}}_{j}$ inside the braces. We also rewrote the surrounding text and added an additional explanatory paragraph immediately after equation (8) to clarify the construction and improve the flow.
>
> **3) Algorithm 1**
>
> - We have revised Algorithm 1 to improve clarity and correctness.
>
> - Specifically, we now explain each input in plain English in the input section, and we have added line numbers to the algorithm for easier reference.
>
> - We also clarified the definition of $H^{\ast}$ and its relationship to $H$.
>
> - In addition, we corrected equation in line (7) of Algorithm 1 by explicitly writing $\gamma \in \Gamma$ under the minimization on the right-hand side to make clear that the optimization is performed over the parameter space $\Gamma$. these changes make the algorithm and its role in the method much easier to follow.
>
> **4) Paper readability**
>
> We have carefully revised the paper to improve readability and to reduce unnecessary measure-theoretic and notational complexity.
>
> - We simplified the notation throughout the paper. In particular, expressions such as $\hat{\boldsymbol{\gamma}}(F^{\mathrm{Pos}})$ are no longer used as subscripts and have been replaced by the simpler notation $\hat{\boldsymbol{\gamma}}$. As a result, Equation (16) now uses a single-level subscript, avoiding nested superscripts and making the expression easier to read.
>
> **5) Clarifying some notations**
>
> - There are, however, a few places where additional simplification is not possible without introducing ambiguity. The following examples illustrate such cases:
>   - In Algorithm 1 we use the notation $L_{\hat{\boldsymbol{\gamma}}^{(i)}}$, where the superscript $(i)$ indicates the $i$-th posterior bootstrap sample. This superscript is necessary to distinguish different bootstrap realizations of the critic parameter. Using a notation such as $\hat{\boldsymbol{\gamma}}(i)$ is not possible here, since $\hat{\boldsymbol{\gamma}}(\cdot)$ is already defined in Equation (14) as a functional of the posterior distribution, and reusing parentheses would create ambiguity. The chosen notation therefore provides a clear and unambiguous way to index bootstrap samples while remaining consistent with the earlier definition.
>
>   - we retained the superscript “DP” in Equation (15) to clearly distinguish the Dirichlet-process–weighted objective from its empirical counterpart. Removing this notation could lead to confusion between objectives using uniform weights and those involving Dirichlet-process weights and atoms, especially given the posterior representation in Equation (7).
>
> - Finally, Section 3.4 has been fully rewritten to provide a much simpler and more intuitive description of the VAE-GAN application. The revised presentation focuses on its main structure, treating network parameters as posterior functionals, to maintain consistency with the BNPL interpretation, while avoiding unnecessary technical detail. This perspective explains why the resulting formulation differs slightly from standard VAE-GAN descriptions. Additional details are provided in Appendix C.
>
> **6) Tone adjustment in empirical results**
>
> - We agree and have revised the text accordingly by removing the term “excellent.” The discussion now focuses on the measurable improvements provided by the DPMINE estimator, without making strong qualitative claims about the reconstruction results.

---

### Author Response · Authors · 2025-12-17
**Public comment to AE and Reviewers**

**Dear Action Editor and Reviewers,**

We sincerely thank the reviewers and the Action Editor for their careful reading and constructive feedback. The comments were extremely helpful and enabled us to substantially strengthen the paper and clarify its focus. We conducted a major revision, with the main changes summarized below:

- **Paper restructuring and focus:**
  The abstract and introduction were completely rewritten, and the entire manuscript was revised to clearly center on the main contribution. Fundamental concepts in Bayesian nonparametric learning and the Bayesian bootstrap are now explicitly introduced, and the proposed method is formulated within a posterior bootstrap framework. The title has been revised to better reflect this contribution.

- **Algorithmic clarity:**
  A clear posterior bootstrap algorithm is now provided, and the connection to existing Bayesian nonparametric learning and Bayesian bootstrap frameworks is explicitly discussed.

- **Experimental redesign:**
  The experimental section has been fully redesigned to better demonstrate the value of the proposed method for neural mutual information estimation. Extensive new experiments have been added to evaluate different aspects of the procedure.

- **Additional toy examples:**
  More challenging toy examples have been added to the Appendix E.1 to illustrate the behavior of the proposed method under complex settings.

- **Benchmark comparisons:**
  Three relevant benchmark methods have been added for comparison, including JSD-LB (Dorent et al., 2025), the most recent state-of-the-art benchmark.

**Sincerely,**
**The Authors**

---

### Decision · Action_Editor_aJux · 2026-02-16

**Recommendation:** Accept as is

**Additional Comments:**

The two rounds of reviews resulted in a (largely) different paper, as compared to the initial submission. The three reviewers now support acceptance of the paper. The main remaining concerns are around the clarity of the writing for readers that are non-expert on BNP.

**Audience:**

Yes

**Audience Explanation:**

The contributions on mutual information will be interested to researchers in mutual information estimation. Additionally, researchers on Bayesian nonparametrics will find this paper of parcticular interest due to the "creative use of BNP ideas" (quoting reviewer sopd).

**Claims And Evidence:**

Yes

**Claims Explanation:**

This paper addresses the problem of estimating the mutual information (MI). It introduces a Bayesian non-parametric MI estimator that uses approximate samples from a Dirichlet process posterior to construct a loss. This offers several advantages for training MI neural estimators (MINE), improving convergence and mitigating sensitivity to outliers. The approach is also demonstrated on deep generative models (especifically, on variational autoencoders), where the MI posterior is used as a regularizer, yielding improved convergence and representation quality.

Overall, after thorough revisions of the submission, all three reviewers find the claims are well supported through a combination of theory and empirical validation. The contributions are now clearly positioned within the related literature.